

# Synchrotron scanning reveals the palaeoneurology of the head-butting *Moschops capensis* (Therapsida, Dinocephalia)

Julien Benoit[1,2,*], Paul R. Manger[2,*], Luke Norton[1,*], Vincent Fernandez[3,*] and Bruce S. Rubidge[1,*]

[1] Evolutionary Institute, School of Geosciences, University of the Witwatersrand, Johannesburg, South Africa
[2] School of Anatomical Sciences, University of the Witwatersrand, Johannesburg, South Africa
[3] Beamline ID19, European Synchrotron Radiation Facility, Grenoble, France
[*] These authors contributed equally to this work.

## ABSTRACT

Dinocephalian therapsids are renowned for their massive, pachyostotic and ornamented skulls adapted for head-to-head fighting during intraspecific combat. Synchrotron scanning of the tapinocephalid *Moschops capensis* reveals, for the first time, numerous anatomical adaptations of the central nervous system related to this combative behaviour. Many neural structures (such as the brain, inner ear and ophthalmic branch of the trigeminal nerve) were completely enclosed and protected by bones, which is unusual for non-mammaliaform therapsids. The nearly complete ossification of the braincase enables precise determination of the brain cavity volume and encephalization quotient, which appears greater than expected for such a large and early herbivore. The practice of head butting is often associated with complex social behaviours and gregariousness in extant species, which are known to influence brain size evolution. Additionally, the plane of the lateral (horizontal) semicircular canal of the bony labyrinth is oriented nearly vertically if the skull is held horizontally, which suggests that the natural position of the head was inclined about 60–65° to the horizontal. This is consistent with the fighting position inferred from osteology, as well as ground-level browsing. Finally, the unusually large parietal tube may have been filled with thick conjunctive tissue to protect the delicate pineal eye from injury sustained during head butting.

Corresponding author
Julien Benoit, julien.benoit@wits.ac.za

# INTRODUCTION

Dinocephalia were middle Permian therapsids that lived some 260–265 million years ago, during the middle Permian (*Rubidge & Sidor, 2001*; *Kemp, 2005*). They were typically large-bodied carnivorous (e.g., Anteosauridae) or herbivorous (e.g., Titanosuchidae, Tapinocephalidae) species, with interdigitating incisors and a pachyostotic (thickened) and ornamented skull (*Rubidge & Sidor, 2001*; *Kemp, 2005*). The taxon name, Dinocephalia, which means 'terrible head', signifies their thickened skull roof bones and pachyostosed

cranial embellishments, such as the fronto-parietal shield (FPS) of *Moschops*, the horns of *Struthiocephalus* and *Estemmenosuchus*, and supraorbital and angular bosses of anteosaurids (*Boonstra, 1936*; *Brink, 1958*; *Olson, 1962*; *Rubidge & Sidor, 2001*; *Kemp, 2005*; *Kammerer, 2011*). The impressive thickness of the cranial vault and the development of pachyosteosclerotic horn-like bosses in tapinocephalid dinocephalians highlight their morphological adaptation for direct and potentially intense head-to-head combat (*Barghusen, 1975*; *Benoit et al., 2016a*). Recognition of the adaptations for head butting in these dinocephalians was not an easy task. Although head butting is a rather common practice among extant ungulates, comparisons with the fossil record are complicated as head butting encompasses a wide variety of behaviours, and because the horns, antlers and bosses of ungulates that are used for fighting are mostly made of keratin, which does not readily fossilize (*Geist, 1966*; *Emlen, 2008*; *Benoit et al., 2016a*). Moreover, unlike dinocephalians, the osseous correlates of ungulates are not made of compact bone, but rather filled with air sinuses (*Geist, 1966*; *Emlen, 2008*; *Farke, 2008*; *Farke, 2010*; *Benoit et al., 2016a*). Cranial fighting surfaces made of compact bone are encountered in odontocetes only, particularly in the maxillary crests of the bottlenose whale (*Hyperoodon ampullatus*) (*Gowans & Rendell, 1999*; *Lambert, Buffrénil & Muizon, 2011*; *Bianucci et al., 2013*). This has resulted in the lack of clear extant analogues to the cranial specialisations of dinocephalians.

More than half a century passed after the name Dinocephalia was coined (*Seeley, 1894*) before *Brink (1958)* and *Barghusen (1975)* hypothesized that dinocephalian cranial features were adaptations for intraspecific head butting contests. *Barghusen (1975)* pointed out that the robust architecture of the skull (with its thickened skull roof, post-orbital bar, and temporal arch) was an adaptation to accommodate direct impacts on the cranial vault. He also argued that the anteroventral position of the occipital condyle (and the resulting anterior position of the quadrate condyle) resulted in a forward inclination of the skull (with the nose pointing downward and the FPS facing forward) that places the neck, the foramen magnum and the FPS in the same plane. This allowed the impact surface of the skull to be aligned with the neck during combat so that energy resulting from blows was transferred and dissipated from the dermatocranium directly to the vertebral column (*Barghusen, 1975*). Supraorbital thickening to absorb mechanical stress in super-carnivorous species as been invoked to explain pachyostosis of this region of the skull in anteosaurids (*Kammerer, 2011*), but at least for tapinocephalids (and for *Moschops* more particularly), Barghusen's morpho-functional reconstruction has convinced most scholars, and the head butting theory is now generally accepted (e.g., *Geist, 1972*; *Carroll, 1988*; *King, 1988*; *Rubidge & Sidor, 2001*; *Kemp, 2005*; *Benton, 2005*; *Benoit et al., 2016a*).

Given the current state of knowledge, it is hypothesized that head butting profoundly altered the cranial osteology of head butting dinocephalians, allowing the cranium to not only physically resist violent impacts that could smash regular bone, but also to protect the delicate central nervous system (CNS) (*Benoit et al., 2016a*). Accordingly, the CNS may have also been modified to withstand these blows and maintain functionality under the same conditions. It may thus be expected that the brain endocast and other osseous structures which reflect CNS morphology, such as the bony labyrinth and cranial nerves, may be highly modified in head butting dinocephalians compared to the usually conservative

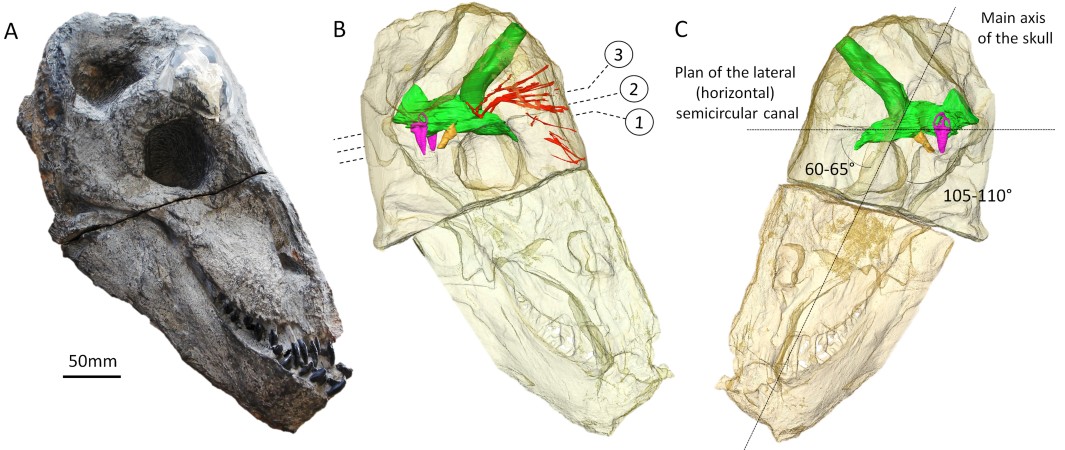

**Figure 1** **The skull of *Moschops capensis* AM4950 in lateral view.** (A) Photograph of the skull. (B) Reconstruction of the skull (right side, bone transparent) to reveal the neural structures discussed in this paper. (C) Reconstruction of the skull (left side, bone transparent) showing the endocast, bony labyrinth and the angle between the plane of the lateral semicircular canal and the main axis of the skull. Numbers indicate the position of the cross sections in the subsequent figures. EmV, emissary veins; End, endocranial cast; Hyp, hypophyseal fossa; Lab, bony labyrinths; Pin, pineal tube. Photo by LN.

CNS morphology in other non-mammalian therapsids (NMT) and Mesozoic mammals (*Olson, 1944*; *Kemp, 1969*; *Kemp, 1979*; *Kemp, 2009*; *Sigogneau, 1970*; *Sigogneau, 1974*; *Jerison, 1973*; *Hopson, 1979*; *Quiroga, 1980*; *Quiroga, 1984*; *Luo, 2001*; *Kielan-Jaworowska, Cifelli & Luo, 2004*; *Rowe, Macrini & Luo, 2011*; *Castanhinha et al., 2013*; *Rodrigues, Ruf & Schultz, 2013a*; *Rodrigues, Ruf & Schultz, 2013b*; *Laaß, 2015*). In addition, it would be of interest for palaeobiological reconstructions to determine how the modified posture of the head and the resulting cranial remodelling (e.g., change in position and orientation of the supraoccipital plate), have affected the positioning of the brain cavity and bony labyrinth relative to the skull. The morphology of the lateral semicircular canal of the bony labyrinth would be especially relevant for this purpose, as it usually remains horizontal with respect to the ground surface when the head is in its 'natural' alert position (*Girard, 1929*; *Giffin, 1989*; *Witmer et al., 2003*; *Witmer et al., 2008*; *Sereno et al., 2007*; *Araujo et al., 2016*). Finally, head butting often involves ritualized display, intimidation ceremonies and other complex, social behaviours (*Geist, 1966*; *Emlen, 2008*), suggesting that head butting dinocephalians expressed significant behavioural complexity (*Geist, 1972*), that may be reflected by the relative size of their brain (*Jerison, 1973*; *Pérez-Barbería & Gordon, 2005*).

In order to explore these hypotheses, here we analysed the very first synchrotron scans ever performed on a dinocephalian, a sub-adult *Moschops capensis* (Fig. 1). *Moschops capensis* is a tapinocephalid dinocephalian from the *Tapinocephalus* Assemblage Zone (late Permian, Wordian, ~265 Ma) of the Beaufort Group of the South African Karoo. This study aims to elucidate potential modifications of the CNS that may be adaptations allowing for combative behaviour in this species.

## MATERIAL AND METHODS

Specimen AM 4950 (Fig. 1) is a nearly complete skull preserved with mandibles. The right side is heavily weathered but the left side is perfectly preserved (except for the supraorbital boss). It was found on the farm The Grant 39 (*Tapinocephalus* Assemblage Zone, late Permian, Wordian, ∼265 Ma), north of Grahamstown, and is housed in the Albany Museum (Grahamstown, South Africa). It was preliminarily assigned to an anteosaurid prior to preparation (*Modesto et al., 2001*), but is now considered to be a sub-adult specimen of *Moschops capensis* (*Benoit et al., 2016a*) based on the lack of a developed canine tooth, presence of talons and heels on all the teeth, broad intertemporal surface, massive postorbital bar, absence of naso-frontal boss, incomplete development of the supraorbital bosses and the unfused state of most cranial sutures, including the bones making up the fronto-parietal shield (FPS). The basal skull length of AM 4950 is 34.02 cm, which indicates that this individual had already reached adult size (basal skull length in adult *Moschops capensis* usually approximates 32–38 cm (*Gregory, 1926*; *Barghusen, 1975*; *Quiroga, 1980*)). Hence, despite the somehow unfused states of many of its cranial sutures, the anatomy of this sub-adult specimen is likely to reflect that of a fully grown adult. This skull was scanned in two parts at the European Synchrotron Radiation Facility (Grenoble France), first the snout in 2007, and then the braincase in 2015. The temporary export of the material for scanning was allowed by the South African Heritage Resources Agency (cases 8090 and 8560).

The anterior portion of the skull of AM4950 was scanned on the ID17 beamline of the ESRF (proposal EC192, 2007) using propagation phase contrast synchrotron radiation microtomography (PPC-SRµCT). The setup consisted of a monochromatic beam at 96.5 keV (single Laue crystal) from a W150 wiggler (gap 34 mm), 5 m of propagation (sample-detector distance), a tapered fiber optic scintillator, an x0.3 magnification set of lenses and a FReLoN-2k14 CCD camera producing data of 45.5 µm voxel size. The tomography was reconstructed based on 5,000 projection of 0.5 s exposure each over 360° with the center or rotation shifted near the edge of recorded radiographs to almost double the horizontal reconstructed field of view. Additionally the fossil was set in a tube filled in with 5 mm Al balls to homogenize attenuation and increase exposure while preventing saturation of the detector (i.e., attenuation protocol). Tomographic correction was done with the program PyHST which generated a stack of 16 bits TIF files. Ring corrections was performed on this stack of TIF files. Finally data was binned to produced an isotropic voxel size of 91 µm to reduce the overall load on computer for segmentation.

The braincase of this skull was scanned at the ID17 beamline of the European Synchrotron Radiation Facility (ESRF, Grenoble, France; proposal ES339). The setup consisted of a FReLoN-2eV camera, a 0.5× magnification set of lenses, a 2 mm LuAG scintillator, a monochromatic X-ray beam of 150 keV (bent double-Laue crystals) and a sample-detector distance of 10.9 m to perform Propagation Phase Contrast Synchrotron micro Computed Tomography (PPC-SRµCT). The tomography was computed based on 2,510 projections (58 × 1,024 pixels, binning factor of 2) of 0.5 s each over 360° resulting in data with a 117.23 µm isotropic voxel size. An attenuation protocol (*Carlson et al., 2001*)
allowed an increase in the exposure time, to compensate for X-ray attenuation by the sample, without saturating the detector. Additionally, the centre of rotation was shifted by ~35 mm to increase the horizontal field of view in the reconstructed data (i.e., half acquisition protocol). Given the limited vertical field of view, 50 scans were necessary (30% of vertical overlap between two consecutive scans) to cover the full height of the sample. Tomographic reconstruction was performed using the single distance phase retrieval approach of the software PyHST2 (*Paganin et al., 2002*; *Mirone et al., 2014*). The resulting 32 bits data were converted to a stack of 16 bit tiffs using the min and max crop values from the 3D histogram generated by PyHST2.

The complete scans of both the snout and cranium can be accessed on the ESRF website: http://paleo.esrf.fr/picture.php?/2296/category/1753.

Three-dimensional renderings of the internal structure of the maxillary canal, bony labyrinth and braincase of *Moschops* were achieved using manual segmentation under Avizo 8 (FEI VSG; Hillsboro OR, USA). All measurements, CT images and 3D rendering were obtained using Avizo 8 (FEI VSG; Hillsboro OR, USA). Following *Benoit, Manger & Rubidge (2016b)*, the branches of the trigeminal canals were named after the corresponding branches of the trigeminal nerve in mammals (see *Benoit, Manger & Rubidge, 2016b* for a discussion).

## Endocast volume, body mass and encephalization quotient

Measurements of endocranial volume were taken using Avizo 8 (FEI VSG; Hillsboro OR, USA). Since the olfactory bulbs are not ossified in AM4950 or in any other NMT, endocast volume was measured excluding the olfactory bulbs and tracts, as was commonly done in earlier studies (e.g., *Jerison, 1973*; *Quiroga, 1980*; *Quiroga, 1984*; *Kielan-Jaworowska, Cifelli & Luo, 2004*; *Rowe, Macrini & Luo, 2011*; *Rodrigues, Ruf & Schultz, 2013a*; *Rodrigues, Ruf & Schultz, 2013b*). The volume of the bony labyrinth was also excluded. Previously published accounts regarding endocranial volume in therapsids included the pineal tube as part of the endocast (*Boonstra, 1968*; *Jerison, 1973*; *Quiroga, 1980*; *Quiroga, 1984*; *Kielan-Jaworowska, Cifelli & Luo, 2004*; *Rowe, Macrini & Luo, 2011*; *Castanhinha et al., 2013*; *Laaß, 2015*). Thus, even though the pineal tube is enormous and contributes a significant portion of the volume of the endocast in *Moschops*, the endocast volume in this study includes the pineal tube in order to allow comparisons with previously published datasets on therapsid endocast volume (a measure of the endocast volume excluding the pineal tube is nevertheless provided for comparison, Table 1).

Published endocast volume data was used to compare the endocast volume of AM4950 to that of other therapsids (Table 1). As endocranial volume cannot be compared directly between species of differing body mass, we used the encephalization quotient (EQ) for comparative purposes (*Jerison, 1973*; *Manger, 2006*; *Hurlburt, Ridgely & Witmer, 2013*). The EQ is the ratio between the endocast volume of a given animal and the expected volume of the endocast for an animal of similar body mass (*Jerison, 1973*; *Manger, 2006*; *Hurlburt, Ridgely & Witmer, 2013*). We used three different types of EQ calculations in our dataset. First, *Jerison*'s (*1973*) EQ, which is calculated as follows:

Jerison's EQ $= (\text{endocast volume in cm}^3)/(0.12*(\text{body mass in g})^{2/3})$.

**Table 1  Measurements of the endocranial cast and calculations of body mass and encephalization quotients in Therapsida.**

| | | Skull length (mm) | BM1 (g) | BM2 (g) | BM3 (g) | Average BM (g) |
|---|---|---|---|---|---|---|
| *Moschops* | Dinocephalia | 340 | 106,299 | 215,918 | 63,748 | 128,655 |
| *Moschops* (no pineal tube) | Dinocephalia | – | – | – | – | 128,655 |
| *Moschops* (*Jerison, 1973*'s BM) | Dinocephalia | – | – | – | – | 327,367 |
| *Moschops* (*Bakker, 1975*'s BM) | Dinocephalia | – | – | – | – | 700,000 |
| *Strutiocephalus* | Dinocephalia | 443 | 234,686 | 493,346 | 136,495 | 288,176 |
| *Strutiocephalus* (*Bakker, 1975*'s BM) | Dinocephalia | – | – | – | – | 1,000,000 |
| *Pristerodon* | Dicynodontia | – | – | – | – | 1,358 |
| *Pristerodon*[*] | Dicynodontia | 78 | 1,281 | 2,149 | 1,012 | 1,481 |
| *Niassodon* | Dicynodontia | – | – | – | – | 491 |
| *Niassodon*[*] | Dicynodontia | 63 | 675 | 1,101 | 570 | 782 |
| *Lystrosaurus* | Dicynodontia | 180 | 15,746 | 29,444 | 10,342 | 18,511 |
| *Tetracynodon* | Therocephalia | 75 | 1,139 | 1,901 | 910 | 1,317 |
| *Brasilitherium* | Cynodontia | – | – | – | – | 99 |
| *Brasilitherium*[*] | Cynodontia | 38 | 149 | 227 | 154 | 176 |
| *Therioherpeton* | Cynodontia | – | – | – | – | 64 |
| *Therioherpeton*[*] | Cynodontia | 31 | 80 | 120 | 93 | 98 |
| *Probainognathus* | Cynodontia | – | – | – | – | 590 |
| *Probainognathus*[*] | Cynodontia | 65 | 741 | 1,215 | 619 | 858 |
| cf. *Probelesodon* | Cynodontia | – | – | – | – | 3,807 |
| cf. *Probelesodon*[*] | Cynodontia | 120 | 4,666 | 8,276 | 3,310 | 5,417 |
| *Exaeretodon* | Cynodontia | – | – | – | – | 46,877 |
| *Exaeretodon*[*] | Cynodontia | 278 | 58,009 | 114,778 | 35,699 | 69,496 |
| *Massetognathus* | Cynodontia | – | – | – | – | 1,865 |
| *Massetognathus*[*] | Cynodontia | 95 | 2,315 | 3,984 | 1,734 | 2,677 |
| *Diademodon* | Cynodontia | – | – | – | – | 50,000 |
| *Diademodon*[*] | Cynodontia | 288 | 64,497 | 128,203 | 39,506 | 77,402 |
| *Diademodon* | Cynodontia | – | – | – | – | 7,000 |
| *Thrinaxodon* | Cynodontia | – | – | – | – | 700 |
| *Thrinaxodon*[*] | Cynodontia | 69 | 873 | 1,440 | 717 | 1,010 |
| *Morganucodon* | Mammaliaformes | – | – | – | – | 51 |
| *Morganucodon*[*] | Mammaliaformes | 26 | 47 | 69 | 61 | 59 |
| *Hadrocodium* | Mammaliaformes | – | – | – | – | 2 |
| *Hadrocodium*[*] | Mammaliaformes | 12 | 5 | 6 | 11 | 7 |

| | EV (g) | Jerison's EQ | Manger's EQ | Hurlburt et al.'s EQ | Source |
|---|---|---|---|---|---|
| *Moschops* | 61,12 | 0,20 | 0,21 | 5,89 | This study |
| *Moschops* (no pineal tube) | 39,85 | 0,13 | 0,14 | 3,84 | This study |
| *Moschops* (*Jerison, 1973*'s BM) | 61,12 | 0,11 | 0,11 | 3,52 | This study |
| *Moschops* (*Bakker, 1975*'s BM) | 61,12 | 0,06 | 0,06 | 2,31 | This study |
| *Strutiocephalus* | 65,00 | 0,12 | 0,13 | 4,01 | *Boonstra (1968)* |
| *Strutiocephalus* (*Bakker, 1975*'s BM) | 65,00 | 0,05 | 0,05 | 2,02 | *Boonstra (1968)* |

**Table 1** (*continued*)

| | EV (g) | Jerison's EQ | Manger's EQ | Hurlburt et al.'s EQ | Source |
|---|---|---|---|---|---|
| *Pristerodon* | 2,18 | 0,15 | 0,21 | 2,61 | *Laaß (2015)* |
| *Pristerodon** | 2,18 | 0,14 | 0,20 | 2,48 | *Laaß (2015)* |
| *Niassodon* | 1,06 | 0,14 | 0,22 | 2,23 | *Castanhinha et al. (2013)* |
| *Niassodon** | 1,06 | 0,10 | 0,15 | 1,72 | *Castanhinha et al. (2013)* |
| *Lystrosaurus* | 8,00 | 0,10 | 0,12 | 2,25 | *Jerison (1973)* |
| *Tetracynodon* | 2,28 | 0,16 | 0,23 | 2,77 | *Sigurdsen et al. (2012) (using graphic double integration)* |
| *Brasilitherium* | 0,38 | 0,15 | 0,25 | 1,93 | *Rodrigues, Ruf & Schultz (2013a)* |
| *Brasilitherium** | 0,38 | 0,10 | 0,16 | 1,40 | *Rodrigues, Ruf & Schultz (2013a)* |
| *Therioherpeton* | 0,36 | 0,19 | 0,32 | 2,33 | *Quiroga (1984)* |
| *Therioherpeton** | 0,36 | 0,14 | 0,24 | 1,84 | *Quiroga (1984)* |
| *Probainognathus* | 1,20 | 0,14 | 0,21 | 2,27 | *Quiroga (1980)* |
| *Probainognathus** | 1,20 | 0,11 | 0,16 | 1,85 | *Quiroga (1980)* |
| cf. *Probelesodon* | 4,33 | 0,15 | 0,20 | 2,92 | *Quiroga (1980)* |
| cf. *Probelesodon** | 4,33 | 0,12 | 0,15 | 2,41 | *Quiroga (1980)* |
| *Exaeretodon* | 19,19 | 0,12 | 0,14 | 3,23 | *Quiroga (1980)* |
| *Exaeretodon** | 19,19 | 0,09 | 0,11 | 2,60 | *Quiroga (1980)* |
| *Massetognathus* | 3,33 | 0,18 | 0,26 | 3,34 | *Quiroga (1980)* |
| *Massetognathus** | 3,33 | 0,14 | 0,20 | 2,73 | *Quiroga (1980)* |
| *Diademodon* | 26,97 | 0,17 | 0,19 | 4,39 | *Rowe, Macrini & Luo (2011)* |
| *Diademodon** | 26,97 | 0,12 | 0,14 | 3,44 | *Rowe, Macrini & Luo (2011)* |
| *Diademodon* | 8,00 | 0,18 | 0,23 | 3,86 | *Jerison (1973)* |
| *Thrinaxodon* | 1,46 | 0,15 | 0,23 | 2,52 | *Rowe, Macrini & Luo (2011)* |
| *Thrinaxodon** | 1,46 | 0,12 | 0,18 | 2,06 | *Rowe, Macrini & Luo (2011)* |
| *Morganucodon* | 0,33 | 0,20 | 0,35 | 2,38 | *Rowe, Macrini & Luo (2011)* |
| *Morganucodon** | 0,33 | 0,18 | 0,31 | 2,20 | *Rowe, Macrini & Luo (2011)* |
| *Hadrocodium* | 0,05 | 0,24 | 0,51 | 1,99 | *Rowe, Macrini & Luo (2011)* |
| *Hadrocodium** | 0,05 | 0,10 | 0,20 | 0,96 | *Rowe, Macrini & Luo (2011)* |

**Notes.**

BM, body mass; EQ, encephalization quotient; EV, endocast volume.

*Indicates taxa for which BM was recalculated (see 'Material and Methods') (Raw data).

This is the most commonly used EQ in the literature, but since Jerison's EQ is often criticised as it is based on a visually, rather than a mathematically, fitted regression line, we also used the regression equation for calculating EQ from *Manger (2006)*. This is considered more accurate as it is defined mathematically. Though they give similar results, Manger's EQ is preferred over Eisenberg's EQ (*Eisenberg, 1981*) because it incorporates more species into the calculation of the regression equation, and excludes outliers such as primates and cetaceans. It is expressed as follows:

Manger's EQ = (endocast volume in cm$^3$)/(0.0535 * (body mass in g)$^{0.7294}$).

Both these EQ calculations were determined to compare relative brain size across mammals. Since early therapsids are known to have endocranial volumes closer to that of extant reptiles than mammals (*Jerison, 1973*; *Quiroga, 1980*; *Quiroga, 1984*; *Kielan-Jaworowska, Cifelli & Luo, 2004*; *Rowe, Macrini & Luo, 2011*; *Rodrigues, Ruf & Schultz,*
_2013a; Rodrigues, Ruf & Schultz, 2013b_), we used a third expression of EQ calculated by _Hurlburt, Ridgely & Witmer (2013)_, which is adapted to the comparison of smaller endocranial volumes. This EQ is calculated as follows:

$$\text{Hurlburt et al.'s EQ} = (\text{endocast volume in cm}^3)/(0,0155*(\text{body mass in g})^{0,553}).$$

This EQ usually gives higher values due to the dataset on which the regression is based. In all cases, the larger the brain relative to the predicted brain size, the larger the EQ.

When available, body mass was taken from the literature (see Table 1). For _Moschops_, _Struthiocephalus_ and _Lystrosaurus_, body mass was calculated based on skull length because post-cranial material for corresponding individuals was not available. We used three different equations from the literature to estimate body mass. The first equation is that of _Quiroga (1980)_ and _Quiroga (1984)_ which was used to compare the relative endocast volume in cynodonts:

$$\text{Body mass}_1(\text{in g}) = 2,7*((\text{skull length in cm})/10)^3.$$

The second was used by _Hu et al. (2005)_ to address the evolution of body size in early mammals:

$$\text{Body mass}_2(\text{in g}) = 10^{3,13*\log(\text{skull length in cm})-5,59}*1,000.$$

The third equation was used by _Castanhinha et al. (2013)_ and consists of an indirect estimation of the femur length based on skull length in order to estimate body mass. This equation is based on the strong correlation that exists between skull length and femur length in NMS (_Sookias, Butler & Benson, 2012_). This correlation allows the estimation of femoral length using skull length based on the regression line of skull length over femoral length provided by _Sookias, Butler & Benson (2012)_. Body mass is then calculated using the estimated femoral length employing the equation of _Campione & Evans (2012)_. The complete expression of the equation summarizing the full process of conversion from skull length to femoral length to body mass is written as follows:

$$\text{Body mass}_3 \ (\text{in g}) = 10^{(2,9307*(\log(0,6908*(\text{skull length in cm})+4,3337))-2,1677)}.$$

Though based on an indirect estimation of femoral length, this equation gives body masses that are consistent with the other two methods (Table 1). We used the average of these three results to obtain a body mass that was then used to calculate the EQs.

The results of this new method of body mass calculation were compared to those of previous studies for specimens for which a skull length and a body mass were available in the literature (Table 1, taxon marked with a *). For those specimens, our method gave estimates that are consistent, yet a little higher, than in previous studies (Table 1). The only exceptions were the dinocephalians. The body masses calculated for _Moschops_ and _Struthiocephalus_ are 129 kg and 288 kg respectively, which appear low compared to the previous estimations given by _Bakker (1975)_. Indeed, _Bakker (1975)_ argued that Moschopid may have weighed around 700 kg and Struthiocephalid around 1,000 kg, but the methods and specimens he used to reach these conclusions are unknown. These make this sole published body mass estimate for these taxa unreliable. Nevertheless, it points out that our

own results may be underestimated. Based on *Jerison*'s (*1973*) equation to estimate body mass in animals with "heavy habitus", which is expressed:

$$\text{Body mass (in g)} = 0.043 * (\text{BodyLength in cm})^{3.03}$$

and using a body length of 186.7 cm as measured from the tip of premaxilla to the posterior tip of ilium by *Gregory (1926)* on the mounted skeleton of the adult *Moschops capensis* AMNH 5552, the estimated body mass of an adult sized *Moschops capensis* should be approximately 327.4 kg. This other estimate is consistent with the possibility that the body masses calculated above for dinocephalians are underestimated. Given that the basal skull length of our specimen of *Moschops* is equivalent to that of an adult specimen, this discrepancy between our estimation and that based on Gregory's specimen might not be explained by the sub-adult age of our specimen. It is more likely the result of a bias induced by the fact that all the equations we used to derive body masses from skull lengths were designed for small sized animals and thus give poor results for large herbivores such as dinocephalians.

As such, in order to take these discrepancies into account in our discussion, the EQs of *Moschops* and *Struthiocephalus* were also calculated using the body masses given by *Bakker (1975)* and that given by Jerison's equation (this last calculation was performed for *Moschops* only since there is no mounted complete skeleton of *Struthiocephalus*).

## DESCRIPTION AND COMPARISONS

For clarity, the orientation of all endocranial structures described here are based on the assumption that the skull was held with an inclination of 60–65° from the horizontal to reflect the hypothesized natural alert posture of the animal (see 'Discussion'). The term endocast is used to describe the internal mould of the brain cavity only.

### Braincase and endocast

In NMT the endocast is incomplete because ossification is limited to the dorsal aspect of the braincase, and the ventral part is ossified only posterior to the pituitary fossa (*Olson, 1944*; *Kemp, 1969*; *Kemp, 1979*; *Kemp, 2009*; *Hopson, 1979*; *Gow, 1986*; *Kielan-Jaworowska, Cifelli & Luo, 2004*; *Rowe, Macrini & Luo, 2011*; *Castanhinha et al., 2013*; *Rodrigues, Ruf & Schultz, 2013a*; *Laaß, 2015*). In some taxa, the sphenethmoid complex (mostly the orbitosphenoid, in addition to the mesethmoid rostrally, the inter-orbital septum ventrally, and the epipterygoid caudally) forms a gutter that partially ossifies ventrally around the forebrain but still leaves the metopic fissure, a wide gap between the sphenethmoid complex and basicranium, unossified (*Boonstra, 1968*; *Hopson, 1979*; *Ivakhnenko, 2008*) (note that the metopic fissure can be reduced in some rubidgeine gorgonopsians and some biarmosuchians too (*Sigogneau, 1970*)). In contrast, the skull of *Moschops* and other tapinocephalids is more robustly built than in most NMT (*Boonstra, 1968*; *Benoit et al., 2016a*). At the level of the braincase, the cranial vault comprises 50–60 mm thick bone divided into a 15–20 mm osteosclerotic surface forming the FPS and about 40 mm of internal cancellous bone (Fig. 2) (*Benoit et al., 2016a*). Unlike the condition in other NMT, in dinocephalians the sphenethmoid complex and basicranium share a suture,

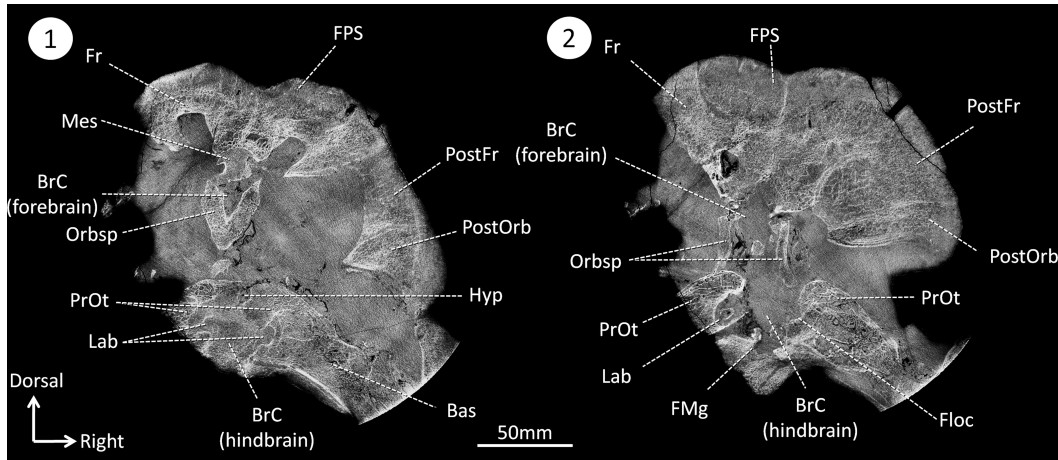

**Figure 2** CT sections of the skull of *Moschops capensis* AM4950 in positions 1 and 2 (see **Fig. 1B**). Abbreviations: Bas, basisphenoid; BrC, braincase; Floc, floccular fossa; FMg, foramen magnum; Hyp, hypophyseal fossa; Mes, mesethmoid; FPS, fronto-parietal shield; Fr, frontal bone; Lab, bony labyrinth; Orbsp, orbitosphenoid; PostFr, postfrontal bone; PostOrb, postorbital bone; PrOt, prootic.

are pachyostotic and cancellous, and the braincase is also completely ossified ventrally (Fig. 2) (*Boonstra, 1968*; *Benoit et al., 2016a*). As a consequence, the endocast is complete in *Moschops*, contrary to that of other NMT.

The endocast volume in this sub-adult specimen of *Moschops capensis* is 61 cm$^3$ (40 cm$^3$ excluding the pineal tube which is 21 cm$^3$), which is close to the 65 cm$^3$ measured in an adult specimen of the more derived tapinocephalid *Struthiocephalus whaitsi* (Table 1). Unlike other NMT (*Kemp, 1969*; *Kemp, 1979*; *Kemp, 2009*; *Hopson, 1979*; *Kielan-Jaworowska, Cifelli & Luo, 2004*; *Rowe, Macrini & Luo, 2011*; *Castanhinha et al., 2013*; *Rodrigues, Ruf & Schultz, 2013a*; *Laaß, 2015*), the main axis of the endocast is not aligned with that of the skull but appears nearly horizontal when the skull is inclined at about 60° from horizontal (Fig. 1C). This inclination of the braincase compared to the rostrum is present to various degrees in all dinocephalians in which the braincase has been studied (*Barghusen, 1975*; *Boonstra, 1968*; *Ivakhnenko, 2008*). The endocast is serially arranged in *Moschops* (Figs. 3A–3C). The cerebral part of the endocast is short and the hemispheres are not distinct on its surface (Figs. 3A and 3B), which means that the cerebral hemispheres were probably small and that the brain was likely separated from the braincase by a thick layer of adnexa, most likely composed of meningeal tissue as well as arteries and venous sinuses (*Bauchot & Stephan, 1967*).

Concerning the olfactory bulbs, only the bony canals that house the olfactory tracts are ossified and are reconstructed here (Figs. 3A–3C). Because of the ventral ossification of the braincase, the endocast of *Moschops* preserves canals for many nerves and other soft tissue structures that are not usually seen on the endocasts of other NMT. For instance, a discrete canal for the optic nerve is present, which is unique to dinocephalians (Fig. 3B) (*Boonstra, 1968*). The metopic fissure is reduced to the passages of the middle cerebral vein and the root of the trigeminal nerve, as in other Dinocephalia (*Boonstra, 1968*). However,

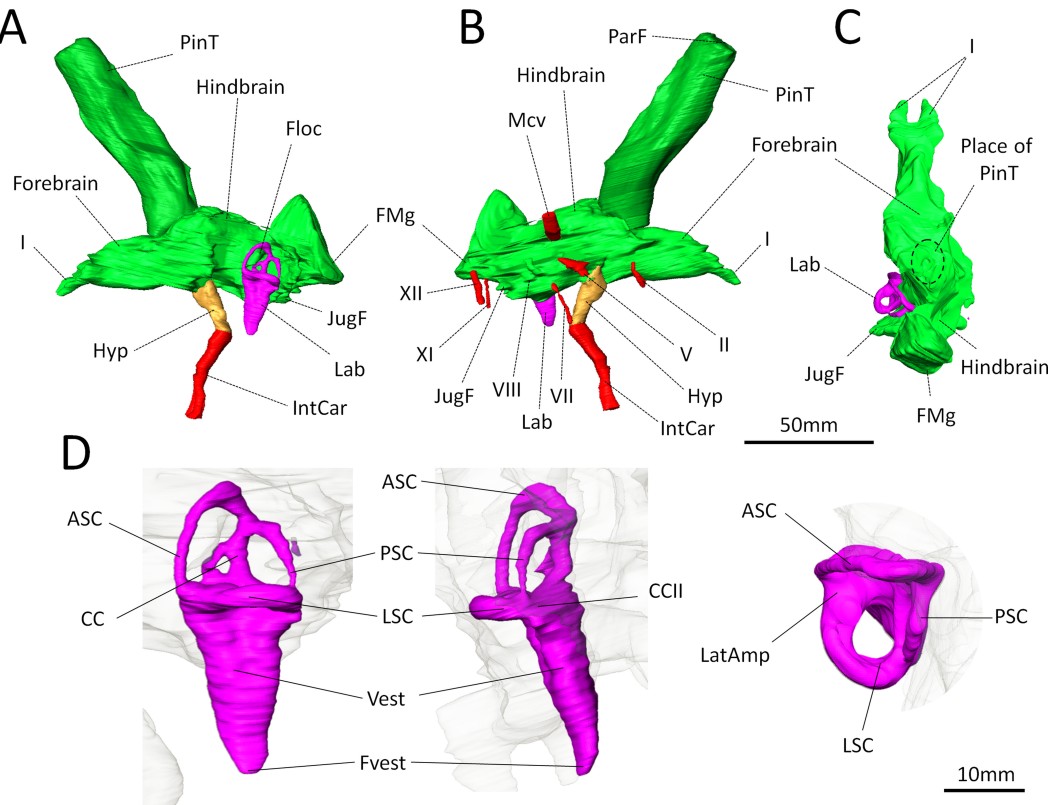

**Figure 3** **Digital reconstruction of the brain endocast and bony labyrinth of *Moschops capensis* AM4950 digitally reconstructed.** (A)–(C), Endocranial cast in lateral (left and right) and dorsal views (pineal tube removed in C for clarity); (D), Bony labyrinth in lateral, posterior and dorsal views. Abbreviations: I, olfactory tract; II, optic nerve; V, trigeminal nerve; VII, facial nerve; VIII, vestibulo-cochlear nerve; XI, accessory nerve; XII, hypoglossal nerve; AntAMp, anterior ampulla; ASC, anterior semicircular canal; CC, common crus; CCII, secondary common crus; FMg, foramen magnum; Floc, floccular fossa; Hyp, hypophyseal fossa; IntCar, internal carotid artery; JugF, jugular foramen; Lab, bony labyrinth; LatAmp, lateral ampulla; LSC, lateral (horizontal) semicircular canal; Mcv, middle cerebral vein; ParF, parietal foramen; PinT, pineal tube; PSC, posterior semicircular canal; Vest, vestibule; Fvest, fenestra vestibuli.

in AM4950 these passages are completely separate, resulting in two distinct canals, a dorsal one for the middle cerebral vein and a ventral one for the root of the trigeminal nerve (Fig. 3B).

A number of large canals, possibly for emissary veins, run from the surface of the FPS to the endocast (Fig. 1B). The pineal tube is prominent in *Moschops* and completely covers the midbrain in dorsal view (Fig. 3C). This tube is oriented slightly rostrally compared to the endocast and the parietal foramen opens right above the midbrain, as in other NMT (*Olson, 1944*; *Kemp, 1969*; *Kemp, 1979*; *Kemp, 2005*; *Kemp, 2009*; *Boonstra, 1968*; *Quay, 1979*; *Hopson, 1979*; *Kielan-Jaworowska, Cifelli & Luo, 2004*; *Castanhinha et al., 2013*; *Laaß, 2015*), but compared to the skull it opens on the caudal margin of the cranial roof (Figs. 1B and 1C). *Moschops* and other dinocephalians are very distinctive because of their hypertrophied and deep hypophyseal fossa (Figs. 3A and 3B) (*Boonstra, 1968*). Unlike other NMT, in dinocephalians the hypophyseal fossa is completely ossified rostrally by

the presphenoid, and its boundaries are well defined (Figs. 3A and 3B) (*Boonstra, 1968*). The hypophyseal fossa occupies a volume of 1.51 cm³. In this sub-adult *Moschops* the hypophyseal fossa is not bulbous, as in the adult *Moschops* and other tapinocephalids described by *Boonstra (1968)*, but it is long and slender, similar to the condition in *Jonkeria* and *Anteosaurus* (Figs. 3A and 3B) (*Boonstra, 1968*). The base of the hypophyseal fossa is pierced by the foramen that transmitted the internal carotid arteries (Figs. 3A and 3B). The stylomastoid canal for the facial nerve is long and located anterior to the bony labyrinth (Fig. 3B). A discrete jugular foramen, for the cochlear canaliculus and the glossopharyngeal and vagus nerves, is found immediately posterior to the bony labyrinth (Fig. 3B). There is a clear osseous separation between the vestibule of the bony labyrinth and the jugular foramen. This is a rare condition amongst NMT, but it has been observed in gorgonopsians and an indeterminated dinocephalian previously (*Haughton, 1918*; *Sigogneau, 1974*; *Luo, 2001*). The canals for the accessory and hypoglossal nerves are separate (Fig. 3B). There is a distinct pontine flexure of the endocast between the hindbrain and the foramen magnum (Figs. 3A and 3B). The floccular fossa is shallow in *Moschops* (Fig. 3A), as in other dinocephalians (*Boonstra, 1968*). It is encircled by the anterior semicircular canal of the bony labyrinth.

## Bony labyrinth

The right side of the braincase is best preserved in AM4950, but the bony labyrinth is complete only on the left side (Figs. 1C and 3D). The vestibule in *Moschops* is long and conical (Fig. 3D) and a small and circular *fenestra vestibuli* opens on its distal extremity (Fig. 3D). There is no evidence for a cochlear recess or canal. The medial ossification of the common crus, anterior ampulla and vestibule is not complete in AM4950, which results in a large unossified area between the endocast and bony labyrinth (Fig. 2). In contrast, adult dinocephalian skulls studied by *Boonstra (1968)* have only a small and discrete internal auditory meatus for the vestibulo-cochlear nerve, which shows that in adult specimens the bony labyrinth and the braincase are separate. The ampullae are inconspicuous in *Moschops* (Fig. 3D), the secondary common crus between the anterior and posterior semicircular canal is long (Fig. 3D), and the anterior semicircular canal appears to be the largest as it projects further dorso-caudally than the posterior canal (Fig. 3D). The lateral semicircular canal forms an angle of about 105–115° with the main axis of the skull (Fig. 1C) which suggests that the actual natural position of the head was more vertical rather than horizontal (see 'Discussion').

## Trigeminal canals

As in other NMT, the trigeminal canals are divided into ophthalmic, maxillary and mandibular branches housing the corresponding rami of the trigeminal nerve and accompanying vessels (Fig. 4) (see *Benoit, Manger & Rubidge, 2016b* for a discussion about the homology of these structures). Although the path of the trigeminal nerve from its root (Fig. 3B) to the trigeminal canals (Fig. 4) is not preserved, the three ramifications are readily identifiable. The maxillary canal is more ramified than that of any other NMT in which it has been documented (*Benoit, Manger & Rubidge, 2016b*); however, it comprises

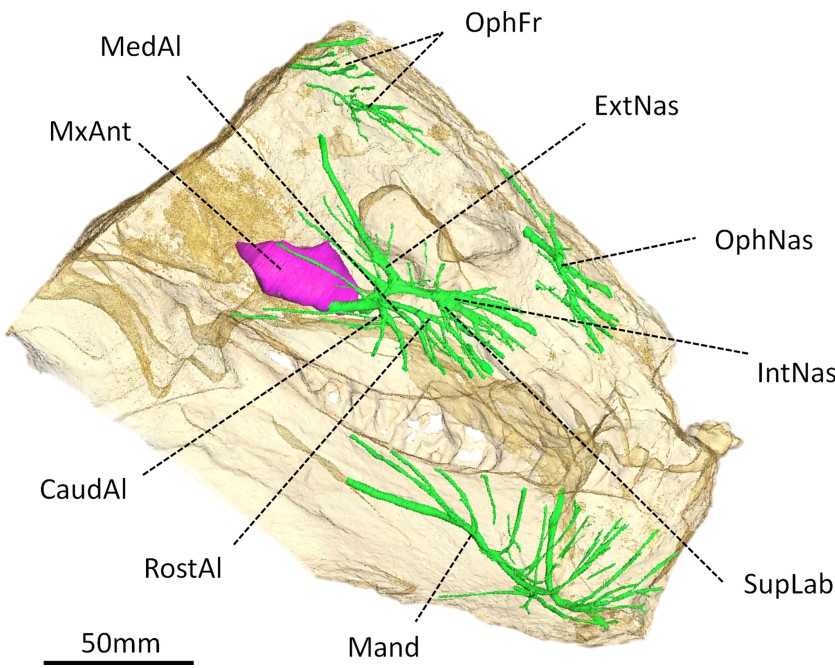

**Figure 4** **Digital reconstruction of the trigeminal canals, presumably for branches of the trigeminal nerve, of *Moschops capensis* AM4950.** Abbreviations: CaudAl, caudal alveolar ramus of the maxillary canal; ExtNas, external nasal ramus of the maxillary canal; IntNas, internal nasal ramus of the maxillary canal; Mand, mandibular ramus; MedAl, medium alveolar ramus of the maxillary canal; MxAnt, maxillary antrum; OphFr, frontal ramus of the ophthalmic branch; OphNas, nasal ramus of the ophthalmic branch; RostAl, rostral alveolar ramus of the maxillary canal; SupLab, superior labial ramus of the maxillary canal.

essentially the same branches as in other taxa. As in most other NMT, three alveolar rami innervate the lip above the maxillary teeth (Fig. 4). These rami are oriented rostrally instead of ventrally, likely in response to the elaborated development of the rostral dentition in tapinocephalids. There is an important postero-dorsal extension of the external nasal ramus, which reaches the bone surface caudal to the external naris (Fig. 4). The internal nasal and superior labial rami reach the rostral-most margins of the maxilla (Fig. 4). The ophthalmic branch is exceptionally well preserved in *Moschops*, with clearly identifiable frontal and nasal rami, which is a rare condition amongst NMT (*Benoit, Manger & Rubidge, 2016b*). Indeed, with the noticeable exception of *Thrinaxodon liorhinus*, the route of the ophthalmic nerve is usually not ossified in NMT and only short and isolated bony channels that go through the nasal and frontal bones mark its presence (*Benoit, Manger & Rubidge, 2016b*). As in *Thrinaxodon*, both the frontal and nasal rami are ossified, but they do not innervate the same area. In *Moschops*, the nasal ramus ramifies inside, and opens on the surface of the premaxilla instead of the nasal bone as in *Thrinaxodon* (Fig. 4). In a similar manner, the frontal ramus ramifies more rostrally than in *Thrinaxodon* and opens on the surface of the nasal bone in *Moschops*, instead of the frontal bone (Fig. 4). Nevertheless, it is possible that some of the canals identified for emissary veins in Fig. 1 could also have carried some branches of the frontal ramus, though it is unlikely given they do not branch

to the canal identified here as the frontal ramus. As for the maxillary canal, the mandibular canal has numerous branches that open into many mental foramina on the surface of the dentary (Fig. 4).

## DISCUSSION

### Bony labyrinth and head posture

The bony labyrinth in *Moschops* is very unusual amongst NMT because compared to other NMT in which all semicircular canals look similar, the bony labyrinth of *Moschops* displays a distinctly larger anterior semicircular canal, a character also commonly encountered in archosaurs (*Hopson, 1979*; *Witmer et al., 2003*; *Witmer et al., 2008*; *Walsh, Luo & Barrett, 2013*) and mammals as well as in the sister taxon of Mammaliaformes, *Brasilitherium* (*Olson, 1944*; *Kemp, 1969*; *Kemp, 1979*; *Kemp, 2009*; *Hopson, 1979*; *Gow, 1986*; *Luo, 2001*; *Kielan-Jaworowska, Cifelli & Luo, 2004*; *Castanhinha et al., 2013*; *Rodrigues, Ruf & Schultz, 2013b*; *Laaß, 2015*; *Ekdale, 2013*). Thirdly, *Moschops* differs from other NMT because the plane of the lateral semicircular canal is not oriented in roughly the same direction as the main axis of the skull (Fig. 1C). As a result, the plane of the lateral semicircular canal is oblique with respect to the horizontal when the skull is held as in most reconstructions (Fig. 5A), whereas a more natural position for the head would be with the plane of the lateral canal closer to the horizontal, parallel to the substrate surface plane ±10° (*Girard, 1929*; *Vidal, Graf & Berthoz, 1986*; *Sereno et al., 2007*; *Witmer et al., 2008*; but see *Marugán-Lobón, Chiappe & Farke, 2013* and *Araujo et al., 2016* for a discussion in the context of NMT). The reconstructed natural alert position of the head is shown in Fig. 5B. According to this new reconstruction, the skull would have been held with the rostrum pointing downward in *Moschops* because in this position, the occiput comfortably articulates with the cervical vertebrae (*Barghusen, 1975*; *Kemp, 2005*). This head posture also positions the FPS facing forward, which is consistent with the reconstructed head butting combat habits of tapinocephalids (*Barghusen, 1975*) and would have facilitated ground-level browsing in these herbivorous animals (*Sereno et al., 2007*). If the head was ordinarily held with the snout lowered down as it is here suggested, this would strongly contradict a semi-aquatic habitat as has been often advocated for tapinocephalids (see *King, 1988* for a review).

As early as 1958, *Brink (1958)* noted that in *Struthiocephalus*, the braincase is tilted posteriorly so that the FPS, the foramen magnum, and the vertebral column were aligned during fighting (*Barghusen, 1975*). This allowed for the transfer of energy to the vertebral column as a result of head butting (Fig. 6) (*Barghusen, 1975*; *Benoit et al., 2016a*). This posterior re-orientation of the braincase is particularly pronounced among tapinocephalids (*Boonstra, 1968*; *Ivakhnenko, 2008*). The tilted condition of the braincase may thus be the by-product of the necessity to align the FPS, foramen magnum and vertebral column during fighting, and also to displace the parietal foramen away from the FPS (*Barghusen, 1975*; *Benoit et al., 2016a*). One may thus argue that the orientation of the bony labyrinth does not reflect the natural head posture, but rather results from this flexure of the basicranium and occiput. However, the re-orientation of the dinocephalian braincase leaves the pontine flexure of the endocast unaffected, meaning that the entire braincase

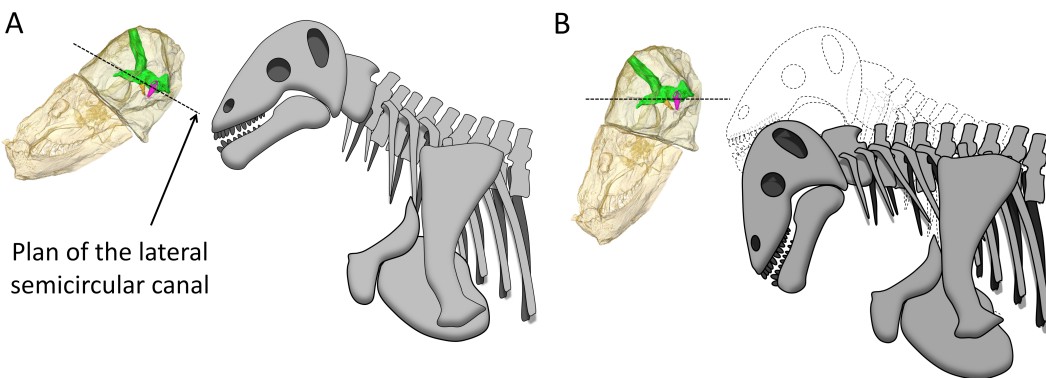

Plan of the lateral
semicircular canal

**Figure 5** **Hypothesized reconstructions of the natural head posture in *Moschops capensis*.** (A) Re-drawn after the mounted skeleton at the American Museum of Natural History (*Gregory, 1926*). (B) Based on the position of the plane of the lateral (horizontal) canal. Artwork by JB.

underwent a re-orientation, not only the basicranium (Fig. 1C) (*Boonstra, 1968*). By comparison, in pachycephalosaurid dinosaurs, the only other reasonably known group displaying a re-organisation of the skull comparable to tapinocephalids, this alignment is achieved by the loss of the pontine flexure (Fig. 7A) which enables the foramen magnum to be positioned right at the base of the skull (*Giffin, 1989*; *Bourke et al., 2014*). However, this leaves the braincase and bony labyrinth in the same orientation as the skull (Fig. 7B) (*Giffin, 1989*; *Bourke et al., 2014*). This indicates that a complete re-orientation of the braincase is not a necessity to align the FPS with the foramen magnum and the neck, thus supporting the hypothesis that the rotation of the bony labyrinth and braincase is not simply an epiphenomenon.

It must be noted here that the backward rotation of the braincase is not unique to dinocephalians as it is also evolved in a convergent manner in some Kannemeyeriiformes such as *Placerias*, *Lystrosaurus* and *Stahleckeria* (Fig. 8; *Camp & Welles, 1956*; *Lehman, 1961*). Interestingly, *Lystrosaurus* displays a strong pontine flexure, like *Moschops*, whereas *Placerias* and *Stahleckeria* do not (Fig. 8; *Camp & Welles, 1956*; *Lehman, 1961*). Unfortunately, the rotation of the braincase (cyptocephaly) is not a very well studied or understood phenomenon in extant species (*Solounias, 2007*), which strongly limits this interpretation. Different degrees of cyptocephaly associated with various degrees of flexure of the pons are documented in many species known to practice head butting, such as artiodactyls (mostly bovids), perissodactyls and proboscideans (*Olson, 1962*; *Olson, 1944*; *Geist, 1966*; *Solounias, 2007*; *Badlangana, Adams & Manger, 2011*). Adaptation for foraging (grazer or browser) is often invoked to account for this pontine flexure, but it remains untested and as such, adaptation to head butting and the effect of phylogeny cannot be ruled out (*Solounias, 2007*). A better understanding of this phenomenon and its ecological implications would bring crucial clues to understanding of the behaviour and palaeoecology of ancient therapsid herbivores.

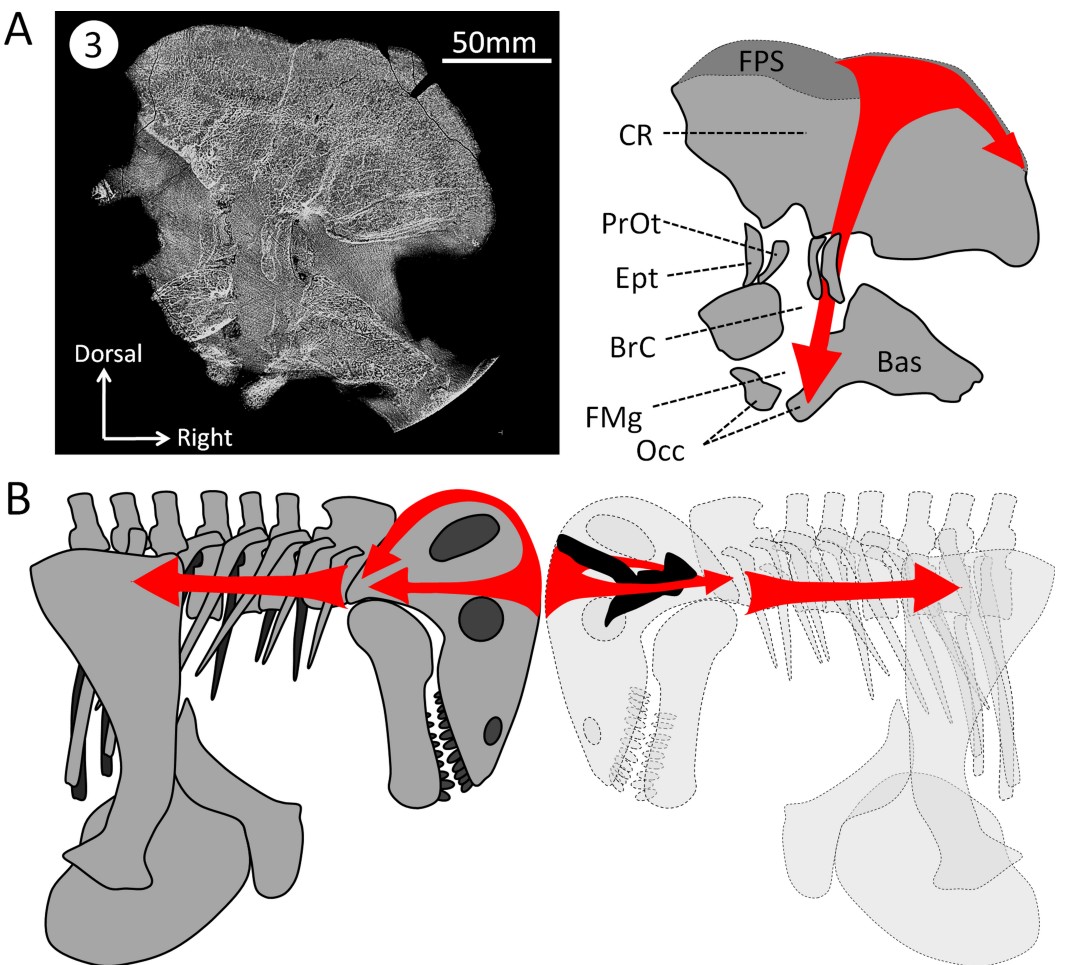

**Figure 6** **Hypothesized dissipation of the energy during head butting in the skull of *Moschops capensis*.** Arrows indicate the direction of energy transfer. (A) CT section of the skull of *Moschops capensis* AM4950 in position 3 (see Fig. 1B). (B) The proposed route of the dissipation of energy through the dermatocranium (left) and the braincase (right) in two fighting *Moschops*. Abbreviations: Bas, basicranium; BrC, braincase; CR, cranial roof; Ept, epipterygoid; FMg, foramen magnum; FPS, fronto-parietal shield; Occ, occipital condyles; PrOt; prootic. Artworks by JB.

## Protection of the CNS and modifications related to head butting

Unlike NMT, the braincase completely encloses and protects the CNS in all mammals (*Hopson, 1979*; *Kielan-Jaworowska, Cifelli & Luo, 2004*; *Rowe, Macrini & Luo, 2011*). Thus, the braincase of extant head butting species resembles that of non-head butting species, though it is often covered by keratinous horn sheaths and filled with enlarged paranasal sinuses that act as shock absorbers (*Farke, 2008*; *Farke, 2010*; *Badlangana, Adams & Manger, 2011*) and the sutures are more ramified and interdigitated (*Jaslow, 1995*). Dinocephalians are unique amongst NMT in having an endocast completely enclosed by the bones making up the braincase (Figs. 2 and 3) (*Boonstra, 1968*). Dorsally, the cranial roof is thickened and covered by a densely ossified FPS, and ventrally the bony complexes of the skull base (sphenethmoid complex and basicranium) are expanded dorsally and medially toward one another to enclose the braincase (Fig. 2) (*Boonstra, 1968*; *Barghusen, 1975*; *Benoit et*

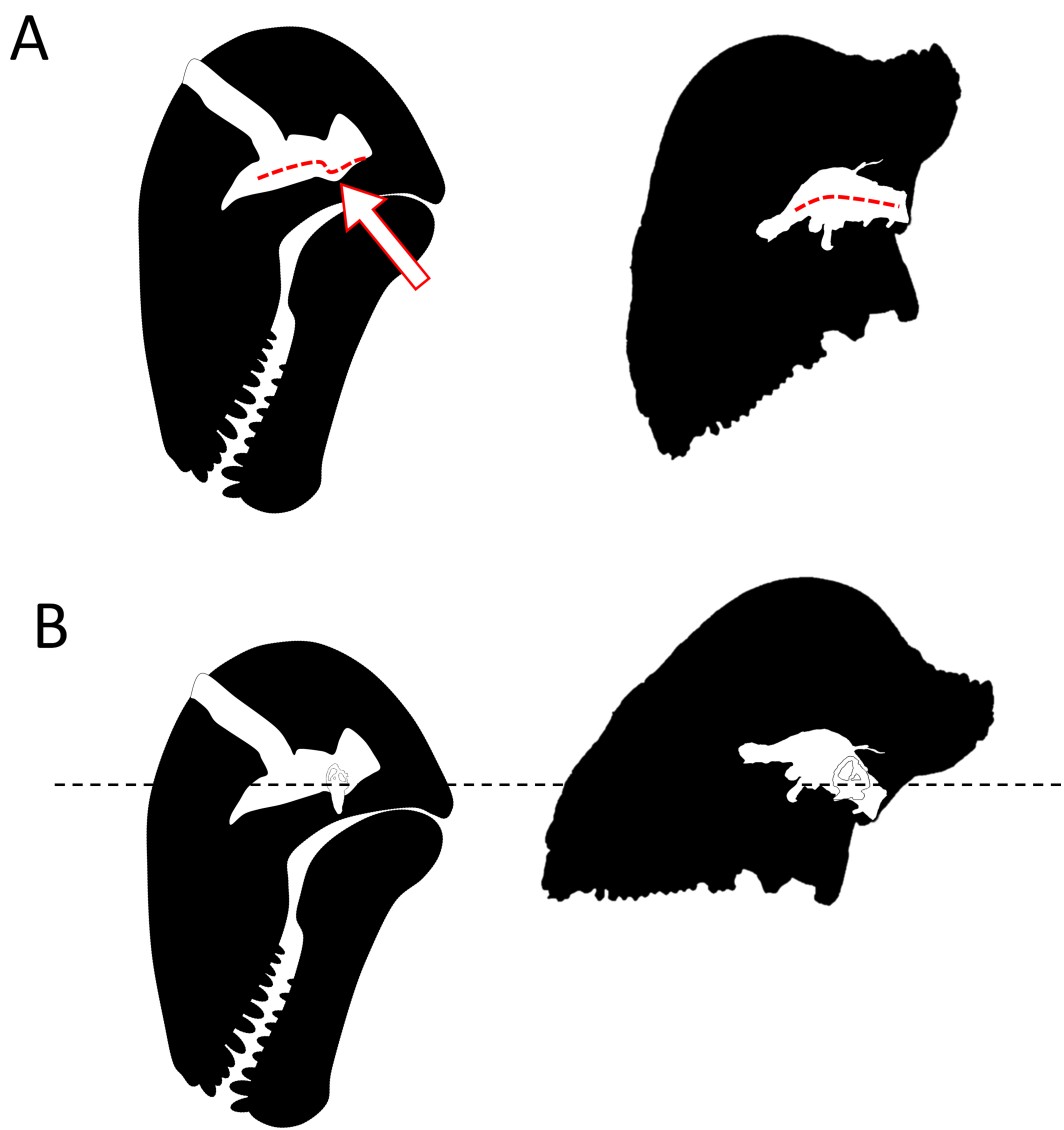

**Figure 7** **Comparison of the orientation of the braincase (white) and the skull (black) in *Moschops* and *Stegoceras*.** (A) comparison of the pontine flexure (indicated by the arrow) in *Moschops* and its absence in the pachycephalosaurid *Stegoceras* (redrawn after *Bourke et al., 2014*). (B) the skull of *Moschops* and *Stegoceras* (redrawn after *Bourke et al., 2014*) aligned according to the plan of their lateral semicircular canals.

*al., 2016a*). Unlike most other NMT, the endocast in *Moschops*, including the hypophyseal fossa, is fully enclosed by bone. The bony labyrinth is isolated from the braincase in adults, and the roots of many cranial nerves are preserved as discrete canals (Fig. 3B) (*Boonstra, 1968*). Only the ventral aspects of the olfactory bulbs are not ossified. Previous studies on head butting in *Moschops* have focused on morphological and histological adaptations such as the thickened bones of the FPS and post-orbital bar, the roughened surface of the FPS to support a cornified plate, the ventral position of the foramen magnum and inclined occiput to align the FPS with neck vertebrae, and the presence of cancellous bones under the dense FPS to absorb energy and lighten the skull (*Barghusen, 1975*; *Benoit et al., 2016a*).

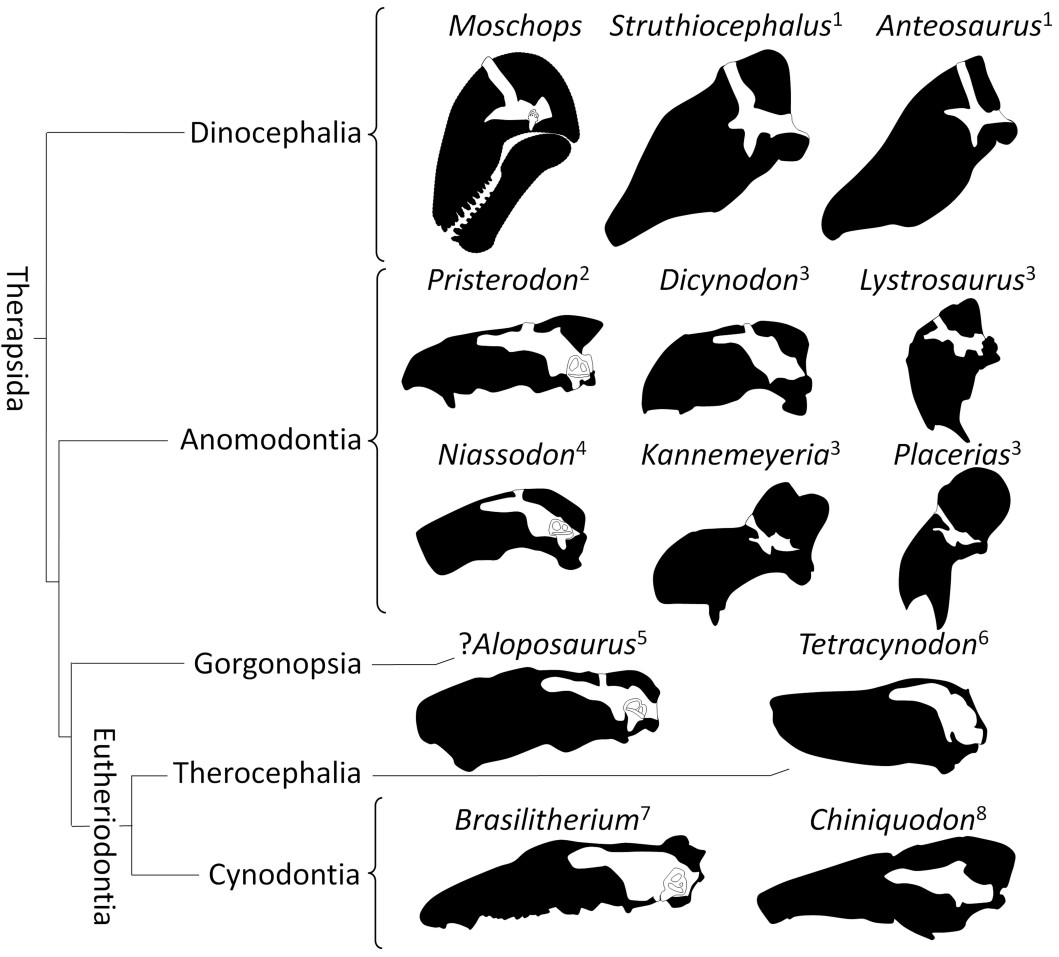

**Figure 8** **Orientation of the braincase (white) compared to that of the skull (black) in Therapsida.** 1, redrawn after *Boonstra (1968)*; 2, redrawn after *Laaß (2015)*; 3, redrawn after *Lehman (1961)*; 4, *Castanhinha et al. (2013)*; 5, *Araujo et al. (2016)*; 6, *Sigurdsen et al. (2012)*; 7, *Rodrigues, Ruf & Schultz (2013a)*; 8, *Kemp (2009)*.

These adaptations prevented bone breakage under physical forces exerted during head butting. The CNS may have also been protected by the thickened orbitosphenoid, prootic and epipterygoid which appear columnar on CT cross section (Fig. 6A). Their anatomy suggest that shock waves were transfered through these bones away from the brain, to the foramen magnum and the neck where they were dissipated (as hypothesized on Fig. 6B). This hypothesis is solely based on comparative anatomy but could be addressed in the future using a functional study based on Finite Element Analysis (FEA) and a broader taxonomic sampling of dinocephalians.

Moreover, since the brain was likely not in direct contact with the braincase in *Moschops*, as indicated by the absence of impressions of the nervous tissue on the endocast (Figs. 3A–3C), they might have been separated by a thick layer of soft tissue (adnexa), presumably meninges, blood vessels (such as the emissary veins), and other conjunctive and protective tissues that surrounded the brain and helped to absorb vibrations.

Adnexa may have occupied much of the pineal tube as well. This tube is comparatively large in *Moschops*, occupying 35% of the volume of the complete endocast (Figs. 3A and 3B; Table 1). The parietal tube and foramen housed the membranous pineal nerve and pineal eye (or third eye) (*Edinger, 1955*; *Quay, 1979*; *Roth, Roth & Hotton, 1986*; *Benoit et al., 2016c*). In extant reptiles, these delicate nervous structures monitor biological cycles and rhythms (*Quay, 1979*; *Roth, Roth & Hotton, 1986*). *Hopson (1979)* argued that the enlarged parietal foramen and the correspondingly large diameter of the pineal tube in dinocephalians might have been for a large pineal eye and nerve. However, in the case of head butting species, the enlarged parietal tube and foramen is more likely to have housed a thick sheath of soft tissue to isolate and protect the pineal eye and nerve from injury during combat. Another possibility is that the large pineal tube was filled with cerebrospinal fluid and operated as a pressure relief window to shunt shocks out of the braincase like the round fenestra act as a pressure relief window for the inner ear (*Luo, Schultz & Ekdale, 2016*).

The complete ossification of a bony tube for the dorsal-most branch of the trigeminal nerve (the ophthalmic branch) may also reflect a protective adaptation related to this part of the CNS. In addition, the ophthalmic branch appears to have shifted rostrally in *Moschops* (Fig. 4), maybe for distancing the ophthalmic nerve (particularly its frontal ramus) and accompanying vessels from the FPS where they could have been injured and triggered pain and bleeding during combat. Therefore, the nasal branch of the ophthalmic canal appears to have shifted to the premaxilla, and the frontal branch appears to have shifted to the nasal bone in order to avoid the surface of the FPS (Fig. 4).

### Endocast volume and behaviour

As dinocephalians display a unique degree of ossification of the braincase, endocast volume can be measured more accurately in this taxon than in other taxa. *Boonstra (1968)* was the first to measure the volume of the endocranial brain cavity in a dinocephalian, a specimen of *Struthiocephalus whaitsi*. Excluding the olfactory bulbs, the endocast of *Struthiocephalus whaitsi* is rather small at only 65 cm$^3$ (excluding the olfactory bulbs), which given its calculated body size (about 288 kg, Table 1), makes its relative endocranial capacity among the smallest according to Jerison's and Manger's EQs (Table 1); however, according to Hurlburt et al.'s EQ, the *Struthiocephalus* endocast is among the largest for NMT (Table 1). Using *Bakker*'s (*1975*) body mass of one ton, the EQs of *Struthiocephalus* are definitely the smallest of all NMT (Table 1), except for Hurlburt et al.'s EQ (Table 1).

With an endocast of about 62 cm$^3$ for a body mass estimated to be about 129 kg, the relative endocranial capacity of *Moschops* is comparatively larger than that of *Struthiocephalus* (Table 1). Depending on the EQ, the relative endocranial size of *Moschops* appears comparable to, or larger than, that of most NMT and early mammaliaforms, such as *Morganucodon* and *Hadrocodium* (according to Jerison's EQ and Hurlburt et al.'s EQ), or it is in the average for NMT (according to Manger's EQ) (Table 1). Using the higher body mass calculated using Jerison's equation (327 kg), the Jerison's and Manger's EQs of *Moschops* still maintain a value of 0.11 (3.52 for Hurlburt et al.'s EQ), which is close to those of some cynodonts and dicynodonts (e.g., *Exaeretodon*, *Lystrosaurus*; Table 1) and is even still comparable to the lowest range of variation of EQs in mammals

(*Jerison, 1973*; *O'Shea & Reep, 1990*; *Manger, 2006*; *Benoit et al., 2013*). This result is quite unexpected for such a large Paleozoic herbivore, given the general trend toward increasing encephalization observed in synapsids (*Hopson, 1979*; *Gow, 1986*; *Kielan-Jaworowska, Cifelli & Luo, 2004*; *Rowe, Macrini & Luo, 2011*; *Castanhinha et al., 2013*; *Rodrigues, Ruf & Schultz, 2013a*; *Laaß, 2015*) and that mammalian herbivores usually have relatively smaller brains and because of negative allometry of brain mass scaling with body mass (*Jerison, 1973*; *Eisenberg, 1981*; *Manger, 2006*). The practice of head butting is often associated with ritual, social and display behaviours, and a hierarchical society in extant gregarious species (*Geist, 1966*). Such complex behaviour, though currently without evidence for Dinocephalia, are known to drive selection for greater brain size in extant ungulates (*Geist, 1972*; *Pérez-Barbería & Gordon, 2005*).

Nevertheless, the endocast volume and EQs of dinocephalians must be interpreted with caution because of the large discrepancies in body mass estimations. Here, using the (probably overestimated) mass of 700 kg advocated by *Bakker (1975)*, the EQs in *Moschops* are unremarkably low (Table 1). In addition, as stated above, the brain did not fill the braincase and thus the endocast volume does not reflect brain size as accurately as it does in mammals. Moreover, the large volume of the pineal tube plays an important role increasing endocast volume as shown by the calculations of EQs where the pineal tube is removed (Table 1). Finally, the hypertrophied hypophyseal fossa also contributes to the volume of the endocast (2.57%, 3.80% when excluding the volume of the pineal tube; Table 1). *Nopcsa (1926)* and *Boonstra (1968)* argued that the large size of the hypophyseal fossa in dinocephalians may reflect the degree of cranial pachyostosis. This would make this character relevant for a discussion about neurological adaptations to head butting, but a similarly hypertrophied hypophyseal fossa is present in many sauropod dinosaurs and other large extinct species in which the skull is not pachyostotic (*Edinger, 1942*). As the hypophyseal fossa houses the pituitary gland, which secretes growth hormones, and since an hypertrophied fossa seems to be present in mostly gigantic species (*Edinger, 1942*; but exceptions exist (*Macrini, Rougier & Rowe, 2007*)), it is more likely that in dinocephalians the size of this fossa may be correlated with large body size rather than cranial pachyostosis.

## Concluding remarks

The skulls of tapinocephalid dinocephalians exhibit extensive adaptations for head butting combat, to the extent that complete ossification and re-orientation of the braincase sets them apart from all other NMT. Intraspecific head butting combat is considered a sexually selected behaviour, but the effect of sexual selection and related behaviours are difficult to define and reconstruct from the fossil record. Therefore the recognition and identification of neural characters that facilitate head butting behaviour in *Moschops* are crucial for future palaeobiological studies of NMT. These characters include the complete bony enclosure of the endocast, bony labyrinth and ophthalmic branch of the trigeminal nerve, a 105–110° inclination of the skull compared to the plan of the lateral semicircular canal, an anterior placement of the ophthalmic canal, and the enlargement of the parietal tube. Given the relatively large number of NMT taxa that manifest signs of head butting behaviour

(*Benoit et al., 2016a*), recognition of such adaptations will certainly change the way the daily life of these long-extinct animals are imagined and also shed new light on the ancestry of mammalian behaviour and sociality.

## ACKNOWLEDGEMENTS

We thank B De Klerk and the Albany Museum for the specimen loan, Dr. R Redelstorff for help with SAHRA regulation and S Jasinoski (ESI) for advice. We acknowledge the European Synchrotron Radiation Facility for provision of synchrotron radiation facilities and we would like to thank P Tafforeau for assistance in using beamline ID17. We also thank the three reviewers, S Lautenschlager, C Kammerer and S Walsh. SAHRA PermitID: 2060.

### Funding

This research was conducted with financial support from PAST (Palaeontological Scientific Trust) and its Scatterlings projects; the National Research Foundation of South Africa; and the DST-NRF Centre of Excellence in Palaeosciences (CoE in Palaeosciences). The funders had no role in study design, data collection and analysis, decision to publish, or preparation of the manuscript.

### Grant Disclosures

The following grant information was disclosed by the authors:
PAST (Palaeontological Scientific Trust).
Scatterlings projects.
National Research Foundation of South Africa.
DST-NRF Centre of Excellence in Palaeosciences (CoE in Palaeosciences).

### Competing Interests

The authors declare there are no competing interests.

### Author Contributions

- Julien Benoit conceived and designed the experiments, performed the experiments, analyzed the data, wrote the paper, prepared figures and/or tables, reviewed drafts of the paper.
- Paul R. Manger conceived and designed the experiments, analyzed the data, wrote the paper, reviewed drafts of the paper.
- Luke Norton performed the experiments, wrote the paper, reviewed drafts of the paper.
- Vincent Fernandez performed the experiments, contributed reagents/materials/analysis tools, wrote the paper, reviewed drafts of the paper.
- Bruce S. Rubidge wrote the paper, reviewed drafts of the paper.

## Field Study Permissions

The following information was supplied relating to field study approvals (i.e., approving body and any reference numbers):

The temporary export of the material for scanning was allowed by the South African Heritage Resources Agency (cases 8090 and 8560).

## Data Availability

The raw data is provided in the Supplementary Materials.

## Supplemental Information

Supplemental information for this article can be found online at http://dx.doi.org/10.7717/peerj.3496#supplemental-information.

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
