# Peer review of "Synchrotron scanning reveals the palaeoneurology of the head-butting Moschops capensis (Therapsida, Dinocephalia)"

_PeerJ, doi:10.7717/peerj.3496_

## Round 0.1 · original submission · Major Revisions

I now have reviews back for your submitted manuscript titled “Synchrotron scanning reveals the palaeoneurology of the head-butting Moschops capensis (Therapsida, Dinocephalia)”. While the three reviewers feel that your manuscript has the potential to become a valuable contribution, they also raise serious concerns. Therefore, my decision is “Major Revision”. If you would like to send a corrected version of your manuscript, please provide a "Response to referees" document in which you list all suggestions and corrections, together with your response to each. I would suggest in particular:

-leaving aside all the speculations mentioned by Reviewer #2 and making it clear how the resulting revised manuscript brings significant new information not available in Benoit et al. (2016; PLoS One);
-improving your figures along the lines of Reviewer #1’s recommendations;
-making the raw data available (e.g., via the ESRF website).

·

Basic reporting

The article is for the major part clearly written and the description of the endocranial anatomy is appropriate. It is clearly structured and a good introduction and background are provided. The latter could be slightly more comprehensive, but this is a minor point and quickly corrected.
Figures are problematic, as they provide too little information or are repetitive (see specific details below).
I suggest moving the supplementary information to the main text, as there is ample space and this will make the manuscript more comprehensive.
Raw data is not provided. Ideally a 3D PDF of the endocranial components, CT data and label fields should be provided, but I realise that this is sometimes not possible due to ownership issues.

Experimental design

The research question is clearly defined and the relevance of the study is clearly stated. Description of methods are OK, but could be more comprehensive. I am a bit doubtful regarding some of the bodymass estimates and these uncertainties should be discussed thoroughly in the manuscript. At the moment they seem to be used to support the author's hypotheses, but have not been questioned.

Validity of the findings

The majority of results are robust. Some of the anatomical structures have been misidentified (see specific comments). The inferences of behaviour and related reorganisation of endocranial anatomy seem logical, but are based on few data points. A thorough comparison with extant analogues would substantiate the findings.

Additional comments

Review of "Synchrotron scanning reveals the palaeoneurology of the head-butting Moschops capensis (Therapsida, Dinocephalia) "


Benoit and colleagues describe the endocranial anatomy of the dinocephalian taxon Moschops and this could be a very valuable contribution as it helps to fill in the blank space of such reconstructions for non-mammalian therapsids. Furthermore, the unusual morphology and behavioural assumptions for this group could allow inferences with regards to endocranial reorganisation in response to social behaviour. However, I feel that the manuscript does not fully live up to this potential. Some anatomical features are incorrectly identified and the lack of comprehensive figures makes it difficult to ascertain certain aspects of the reconstruction. The calculations of the encephalisation quotient and the inferences of complex social behaviour are somewhat doubtful. A more detailed and critical discussion of this and expanded comparisons with other extant and extinct analogues for head-butting behaviour are in my opinion necessary.
That said, I see the potential for a good paper, if these changes are implemented. More detailed comments and specific point are listed below.




- the introductions needs a few sentences on Moschops (occurrence, morphology, phylogenetic context) to provide the reader with the necessary background on the studied taxon
- there is no need for a separate supplementary text. The supplement is actually very short and should really be part of the main text
- the methods section needs more details on the preservation of the specimen and the respective resolution of the CT data set. Were attenuation levels of bone, matrix and structures of interest sufficiently distinct to allow the recognition of structures? Which segmentation tools (paintbrush, magic wand, interpolation, etc) were used in Avizo?
- It would be helpful to provide one or more additional figures of the endocast, endocast orientation and endosseous labyrinth of comparative taxa (possibly plotted in a phylogenetic context) to supplement the described comparisons (e.g. page 10, lines 137-139)
- the description/interpretation of the endosseous labyrinth is not correct (page 12, lines179-181, fig 3D). The structures described as the vestibule is actually the cochlea or cochlear duct. Also the fenestra vestibuli is of located at the ventral tip of the cochlea, but should project perpendicular to the dorsoventral axis below the lateral semicircular canal. Further, the endosseous labyrinth is not "fused" to the endocast (lines 182-183) - consider rephrasing. Is there actually a secondary common crus present, which should connect the posterior and lateral semicircular canals (not as stated the anterior and posterior canals). The figure is very small and unclear regarding this feature and I can't see a secondary common crus.
- the assumption that the brain and related structures undergo a reorganisation in response to head butting seems tempting, but I would like to see a bit more data (or lacking that more discussion) with regards to extant ungulates and other taxa. It is briefly touched upon in lines 261-262, but at the moment this is a bit too arm-wavy to me. I find this phenomenon very interesting and the manuscript would definitely benefit from a more detailed discussion.in particular as the comparison with pachycephalosaurs contradicts the authors.
- I am very sceptical with regards to the large EQ in Moschops. It seems to me that this is an artefact of the body mass estimate, which with 129 kg appears very low for an animal of that size and proportions. All body mass estimates are based on skull length and do not correct for the fact that the postcranial skeleton is very exaggerated in comparison to cynodonts and early mammals (for which Quiroga's and Hu's estimates were used originally). The indirect measurement of femur length is similarly based on skull length and therefore also misleading for the same reason. As shown by Christiansen and Farina (2006), skull length is not a reliable proxy for body mass in highly modified herbivores. I do understand that the authors wasn't to use a comparison for EQs across a range of taxa, but the results should be treated very cautiously and requires more discussion. Instead of using a single value a range for the EQ might be more appropriate to reflect the uncertainties. Also, the authors could calculate body mass for the other taxa in table 1 and compare the resultant EQs with published values. This way all the measurements would be based on the same method for calculating body mass
- the figures could be very much improved. Although the individual figures are generally good, they provide very little information. Figure 1B is used four times but does not show anything new each time, whereas the actual endocranial structures are reduced to same images in only few orientations. It appears to me that the authors have created the figures for a different journal originally and did not make use of the liberal figure allowances of Peerj. At the very minimum a separate figure with the endocranial components in larger scale, more orientations and clearer labels is required. I would further recommend a 3D PDF for the supplement and possibly a deposition of the CT data If possible.

Page 6, line 32: remove comma after "Dinocephalia"

Page 6, line 33: give brief definition of "pachyostotic" as readers might not necessarily be familiar with this term.

Page 6 line 42: replace "ferocious" with "intense" or similar neutral term

Page 6, line 43: "...adaptations for head butting..."

Page 6, line 46: "......horns, antlers and bosses of ungulates..."

Page 6, line 52: remove "...but this was discovered only recently"

Page 7, line 56: "...intraspecific head butting contests..."; also use either "head butting" or head-butting" consistently throughout the text

Page 7, lines 62-67: this requires more explanation; also split into two or three sentences. Currently this sentence is difficult to read

Page 10, line 141-143: it could also mean that the cerebral hemispheres are only weakly developed, thus not "sticking out" through the meninges.

Page 13, lines 193-194: "...branches housing the branches..." - better rephrase

Page 13, line 196: what is this identification based on? The location of the canals and if so, is there phylogenetic support for it?. Needs a bit more explanation and justification. In particular as further down in the text, a significantly different pattern of innervation is described in comparison to Thrinaxodon

Page 14, line 213: "...it is possible that..."
Page 14, lines 221-223: the semicircular canals do not appear particularly thick or robust to me. Can this be quantified more accurately (measurements for canal diameter and radius) and put in context to other taxa? Also the authors state that the ampullae are incompletely preserved, so the statement about their absence might be a taphonomic artefact.

Page 14, line 231-233: see comment above on the correct identification of cochlea

Page 14, line 233-235: this sounds circular. The habitual position of the head is usually inferred based on the orientation of the lateral semicircular canal and not vice versa. Best to rephrase this sentence.

Page 16, line 270: better "...an endocast completely enclosed by the braincase bones..."

Page 18, lines 306-409: as presented this makes no sense. If the nasal branch is diverted to the premaxilla to avoid exposure during head butting, the replacement by the frontal branch just substitutes this problem.

Page 18, line 312: "...more accurately than in other taxa..."

Figure 2: needs orientation of the CT slices. It is difficult to recognise where the anterior and posterior ends are. The position as shown in Figure 1 is also very confusing. Are these sagittal section or coronal? The individual slices suggest that they are sagittal sections, but the indices in Figure 1 only make sense for coronal sections.

·

Basic reporting

Vertebrate paleontology as a science is frequently derided (usually by invertebrate paleontologists) as being heavy on speculation, light on data. Much of this has to do with the desire to understand paleobiology in vertebrate fossil taxa—the incredible body sizes and extravagant cranial structures of Mesozoic dinosaurs and Tertiary mammals, for instance, beg for interpretation in a way that most bivalve shells or foraminiferan tests do not. And it is true that, historically, a lot of ink was spilled trying to interpret such structures in ways that were not exactly subject to rigorous analysis. However, I would argue that such stereotypes are a thing of the past: nowadays vert paleo is as rigorous as any other science, and while we have not abandoned the desire to study paleobiology, modern vertebrate paleontologists publish paleobiological interpretations backed by hard evidence, not baseless speculation.

It is with this in mind that I must express my disappointment with the current manuscript, which provides much in the way of baseless speculation and little in the way of hard data. I will first note that this version of the manuscript is indeed a significant improvement from an earlier version that I had reviewed previously. Most of the basic factual errors have been corrected, and the manuscript is much better-written (although the current version still requires heavy editing—wayward commas run throughout the current text). However, my main complaint with the manuscript as presented stands: this paper is filled with paleobiological claims and interpretations without any data to support them. Or rather, there are two competing papers within this manuscript: part one, a straightforward and very useful morphological description of the braincase of Moschops (that however covers some of the same ground as the previous paper about this specimen: Benoit et al. 2016 in PLoS ONE), and part two, rampant speculation about Moschops behavior and soft-tissue anatomy that is “inspired” but not demonstrated by part one. I will review the morphological description below, but first would like to note all the statements in this manuscript that I would classify as baseless speculation:

Lines 21–24: “The presence of a correspondingly large brain would be consistent with the practice of complex social behaviours such as hierarchical ranking, ritualized display or intimidation, which are often observed in extant head-butting species.” Speculation. Relative brain size in extant mammals may be larger in social species, but is it larger in species that engage in intraspecific combat? Need data or references to back this up.

Lines 27–29: “Finally, the very large parietal tube must have been filled with thick conjunctive tissue to protect the delicate pineal eye from injury sustained during head butting.” Note usage of the word “must” in this statement, which represents complete speculation. Do you know what organs are substantially more delicate and injury-prone than pineal eyes (which are typically embedded at least partially in skin and surrounded by the parietal bones)? Regular eyes, which we know from direct observation often have enormous, pointy horns directed towards them at high speeds by extant head-butting ungulates. And yet there is not a “thick conjunctive tissue” between the bony orbit and eyeball of these animals.

Lines 40–42: “The impressive thickness of the cranial vault and the development of pachyosteosclerotic horn-like bosses in tapinocephalid dinocephalians highlight their morphological adaptation for direct and potentially ferocious head-to-head combat.” Kernel of information here wrapped in lots of speculation, with a problem that runs throughout this whole paper: i.e., starting from the conclusion that dinocephalian cranial structure was for combat (rather than display or something functional) rather than mounting evidence to support that conclusion. Further speculation: that combat was “ferocious” (a perhaps admissible superlative to sex things up) and that it was head-to-head (rather than flank-butting, a more common form of combat among extant amniotes).

Lines 64–66: “Barghusen’s pioneering morpho-functional reconstruction has convinced most scholars, and the head-butting theory is now generally accepted.” OK, this is not speculation but still needs to be addressed here—Barghusen (1975) is a fine piece of work, but most synapsid researchers maintain a healthy air of suspicion about its conclusions. Where Barghusen excelled was making detailed gross morphological comparisons between tapinocephalids and extant ungulates known to practice head-butting, establishing a testable hypothesis. We, the synapsid research community, are still waiting for that hypothesis to be tested. There are several ways to do this, all of which have already been done for other clades by Cenozoic mammal and dinosaur workers. The most obvious of these would be surveys of cranial pathology (is there evidence of consistent lesions on dinocephalian skulls that you would expect from sustained head-butting damage?), FEA analysis (is the frontoparietal dome able to disperse heavy stresses?), or comparison of endocranial morphology with modern head-butters. Of these possibilities at least the last two are eminently possible with the data at your command (honestly the first one is as well given how few dinocephalian skulls there really are and the fact that the vast majority of them are either in Cape Town or Johannesburg, but I digress). We are still waiting for the paper that tests this hypothesis, but you could easily do one of these tests! This paper here, with the addition of comparative data from a mammal, even if it was only one head-butting mammal, could be it! What a wonderful, heavily-cited paper that would be, if we finally had good support for head-butting in dinocephalians, rather than just assuming that was the case a priori.

Lines 68-70: “Head butting profoundly altered the cranial osteology of tapinocephalids, allowing the cranium to not only physically resist violent impacts that could smash regular bone, but also to protect the delicate central nervous system.” Pure speculation. No evolutionary context whatsoever provided for this statement (in either this or the previous paper). How do you know when head-butting appeared in tapinocephalian phylogeny? How do you know that the cranial osteology changed as a result of head-butting behavior? How do you know that, even if Moschops was a head-butter, pachyostosis in earlier tapinocephalids evolved for combat? Where is the comparative anatomy, where is a cladogram, where is the data?

Lines 70–72: “Accordingly, the CNS must have also been modified to withstand these blows and maintain functionality under the same conditions.” Speculation. Is the CNS of extant head-butting ungulates very different from non-head-butting confamilials? Maybe, maybe not, I don’t know! I’m sure that data is out there, you need data to back up your statements. Otherwise it is just hot air.

Lines 72–75: “It may thus be expected that the endocast and other osseous structures which reflect CNS morphology, such as the bony labyrinth and cranial nerves, may be highly modified in tapinocephalids compared to the usually conservative CNS morphology in other non-mammalian therapsids.” Speculation and contradiction—you state later in the text that the endocranium of Moschops differs from other therapsids (in which this region has been studied, which is a distinct minority of known taxa) primarily in being completely ossified. This allows the brain endocast of Moschops to be completely reconstructed with confidence, whereas the mostly cartilaginous endocranium of other therapsids means much of the CNS morphology is unknown. If we don’t know what the CNS was like in other therapsids, how do you know it was different from that of Moschops?

Lines 78–79: “In addition, since tapinocephalid dinocephalians underwent considerable cranial remodeling to accommodate the new posture of the head in order to transfer impact energy to the neck…” Speculation. Have you compared head posture and cervical morphology between tapinocephalid dinocephalians and their sister-taxa (deuterosaurids and titanosuchids)? How do you know they are different? How do you know they changed? (Neither of the references for this statement address this issue.) Comparative anatomy and phylogenetic context are necessary.

Lines 82–85: “Finally, head butting often involves ritualized display, intimidating ceremonies and other complex, social behaviors which would imply that tapinocephalids expressed significant behavioral complexity. If true, this should be reflected in the relative size of their brain.” Speculation. Head-butting is also present in various fish and insects, do they have notably expanded brains? Or rather, does having a tiny brain prevent this behavior (it would seem not)?

Lines 251–253: “The tilted condition of the braincase may thus be the by-product of the necessity to align the FPS, foramen magnum and vertebral column during fighting, and also to displace the parietal foramen away from the FPS.” Speculation. Strongly tilted braincases are also present in some anteosaurids and kannemeyeriiforms among therapsids, were these also head-butters? Difficult to believe considering the tall, thin sagittal crests of kannemeyeriiforms.

Lines 303–306: “Moreover, the ophthalmic branch appears to have shifted rostrally in Moschops, maybe for distancing the ophthalmic nerve and accompanying blood vessels from the FPS where they could have been injured and triggered pain and bleeding during combat.” Speculation.

Lines 328–329: “Alternatively, the practice of head butting is often associated with ritual, social and display behaviours, and a hierarchical society in extant gregarious species.” This may be true, but does not make sense as an explanation for the relatively large (compared to Struthiocephalus) brain of Moschops. Struthiocephalus shows all of the same cranial adaptations (at least as can be seen without CT data) as Moschops, except it has an even more extreme morphology with a rounded boss in the middle of the frontoparietal dome. If anything Struthiocephalus is an even better candidate for head-based combat, why would it differ from Moschops in this regard?

Lines 359–361: “Therefore the recognition and identification of neural characters that facilitate head-butting behavior in Moschops are crucial for future palaeobiological studies of NMT.” What about the recognition and identification of neural characters that facilitate head-butting behavior in extant, known head-butting mammals? Shouldn’t we establish that first, and then intepret fossils using comparative anatomy? So many of my problems with this manuscript would instantly disappear if the authors brought in this single additional data point (or indicated they have read existing literature that accomplishes this goal.) It is frustrating, because the authors are so close to a good and very important paper on this topic but seem to doggedly avoid the crucial information for sealing the deal.

Experimental design

See previous section.

Validity of the findings

In addition to the major concerns I listed in the "Basic Reporting" above, I also have a set of more minor comments and complaints detailed below:

Line 33: I really wish you would use the better-known term “non-mammalian therapsid” rather than “non-mammaliaform therapsid”. Unless the statement you are making necessitates distinction of the Morganucodon grade from more rootward therapsids, there is no reason to specify “non-mammaliaform”.

Lines 35–36: Dinocephalia is an apt name for this clade, but Seeley probably did not create this name to refer to the thickened skull of tapinocephalids. Like Seeley’s other higher taxa, Dinocephalia was named for its “type genus”, in this case Dinocephalus (e.g., Gorgonopsia from Gorgonops, Kistecephalia from Kistecephalus). Unfortunately Dinocephalus remains a nomen nudum and no subsequent papers reveal what Seeley meant this taxon to be (in his 1894 paper he says only that Dinocephalus has “the largest canine teeth found in any South African fossil, associated with small molars”). As such, and given the importance Seeley places on heterodonty in the group in the rest of his description, it seems the name Dinocephalia actually refers to the teeth of these animals, not the skull. This is further supported by his (1894) tentative inclusion of Aelurosaurus (a small gorgonopsian completely lacking pachyostosis, but with exaggerated fangs like all early theriodonts) in Dinocephalia, and the fact that among the two tapinocephalids he listed as dinocephalian (Delphinognathus and Tapinocephalus), the holotype of Delphinognathus represents a weakly-pachyostosed juvenile and the holotype of Tapinocephalus was distinguished mostly by snout morphology. At no point does Seeley (1894) mention thickening of the skull in his description of Dinocephalia, and it is also important to note that he did not include the Russian Deuterosaurus (now known to be a close relative of tapinocephalids) in Dinocephalia, but rather accorded it its own group Deuterosauria.

Lines 46-47: “the horns, antlers and bosses that are used for fighting are mostly made of keratin which does not readily fossilize”—while the horn surface itself may not fossilize, the bony horn core (in bovids or antilocaprids, or ossicones in giraffids and antlers in cervids) most certainly does fossilize and indeed is the most commonly-identified element in the fossil record of these groups.

Line 50: Worth noting that even if only Hyperoodon specifically butts heads, odontocetes in general widely utilized rostral ramming in both intra- and interspecific agonistic behaviors.

Line 86: Are they really new synchrotron scans? Aren’t these the same synchrotron scans already described for this specimen by Benoit et al. (2016)?

Lines 95–99: These characters apply to multiple tapinocephalid genera, this is not a robust diagnosis for Moschops (although I do agree with that identification for this specimen.)

Line 121: rubidgeine is misspelled here

Line 126: “unlike the condition in other NMT…”—unlike other scanned/sampled therapsids, you mean? Because I suspect Tapinocephalus, Ulemosaurus etc. probably had a very similar condition to that of Moschops.

Line 228: either decapitalize mammaliaforms or it should be the clade Mammaliaformes

Line 258: Only other group? What about Triopticus?

Lines 317–318: It would be useful to explain how these EQ measures differ and why they have such contradictory results.

·

Basic reporting

The article passes the four criteria, although the raw data are actually not included - this is probably impractical due to the file sizes of synchrotron tomograph stacks. A 3D PDF or mesh model files of the segmented data would be useful as supplementary materials. Otherwise, minor comments can be found in my 'general comments to authors'.

Experimental design

No comment.

Validity of the findings

No comment, but see my 'general comments to authors' for minor issues.

Additional comments

This MS is a nice succinct description of a study of the skull of the dinocephalian Moschops using a synchrotron dataset. The results broadly support existing ideas about intraspecific head-butting behaviour, but add new information about how CNS architecture is modified in therapsids that appear to be adapted for such behaviour, compared with those that were not. Regardless of the importance of impact to PeerJ, the study has an audience wider than mammal workers only: adaptations to mitigate routine cranial impact damage to the CNS and associated sensory structures are found in other non-mammalian groups such as archosaurs (pachycephalosaurs, woodpeckers). The MS represents a useful comparison for those taxa, and a worthwhile addition to our knowledge of endocranial structures. It certainly should be published.

I have no serious criticisms of the MS in terms of the scientific approach, nor for the most part with the conclusions the authors have drawn from their findings. There are a few points that I think should be addressed, however.

1) There needs to be a better justification for the reliability of using a sub-adult individual. This does not need much in the way of additional text, just a sentence or two explaining why the findings are reliable despite the individual under study not having achieved full adulthood. There are already comments about this throughout the text (e.g. lns 326-8), but the uncertainty weakens conclusions drawn from the data.

2) The introduction (lns 78-82) mentions that the labyrinth and brain cavity may have undergone reorientation in response to impact forces. This statement could do with some explanation as to why this might be necessary (e.g., force distribution more effective if the main brain axis is aligned with the shock propagation pathway; ‘alert’ head position orientation of the labyrinth).

3) A propos labyrinth indexing the alert head position, I have no issues with using the ‘horizontal’ SSC to index this, but the work of Marugán-Lobón et al. (2013 – oddly enough in PeerJ: DOI 10.7717/peerj.124) on this subject has not been cited or discussed. This needs to be included.

4) Are the authors completely confident in their identification of parts of the endocast? Labels such as “hindbrain” tend to suggest the component parts of that general region are difficult to determine (as expected), even though a cerebellar flocculus appears to be visible. If I were looking at an archosaur endocast in Figure 3, I would have interpreted the structure labelled as the hypophyseal fossa as the rami of the internal carotid arteries combined into a single canal (as in many birds), with the swelling at the dorsal end as the fossa that held the hypophysis itself. Likewise, the twin canals labelled as the olfactory tract appear to me to be the foramina of CNI rather than the 'tract'. The elongated ‘vestibule’ looks pretty much like the sort of cochlear canal observed on birds, crocs and vocalising squamates. Are the authors sure this is not the cochlear canal? Can the position of the fenestra ovalis be determined?

5) I have made some other minor comments that the authors may wish to consider on my copy of the MS and supplementary materials, and also pointed out some improvements to the text (especially punctuation). The formatting for Table 1 seems out, and the order of the references is also out in some places. Lastly, make sure references to ‘endocranial measurements’, ‘endocranial casts’ etc. are properly specified. An endocranial cast/endocast could be of any enclosed space within the endocranium, and measurements could be literally of anything in that region! The “endocast” referred to is actually a brain cavity endocast.

---

## Round 0.2 · Minor Revisions

Thank you for submitting the revised version of your manuscript. With your next submission, please provide a "Response to referees" document in which you list the issues raised in this latest round of reviews and how you addressed them. Specifically, you should address "Fig. 6 and the respective discussion of it" (Reviewer #1) as well as the "broader anatomical or phylogenetic context" (Reviewer #2).

I noted that one of the 3D pdfs is no working. Finally, as per PeerJ policies (https://peerj.com/about/policies-and-procedures/#data-materials-sharing) I am afraid all the raw data will have to be made available in a permanent public repository prior to formal acceptance.

The three reviewers made an excellent job, which you might want to acknowledge.

·

Basic reporting

I find the manuscript very much improved compared to the previous version. The methods section has been expanded or moved from the supplement making the manuscript more comprehensive and many speculative paragraphs have been rephrased. The 3D PDF do not work for me and apparently some of the coauthors, so this issue needs to be resolved. Otherwise, all points raised in the initial review have been addressed satisfactorily.

Experimental design

The methods section has been expanded, making the study design more comprehensible. Issue previously raised regarding body mass estimates have been extensively addressed and are now discussed in detail.

Validity of the findings

I find that the majority of findings is robust and many speculations have been removed or rephrased. Some results remain (necessarily) speculative, but are now discussed in more detail and the comparison with other taxa is very helpful. The only objection I have is with Fig. 6 and the respective discussion of it. The hypothesised distribution of stress is based purely on the morphological arrangement and it should be clearly mentioned in the figure caption and the discussion, that this is an untested assumption of the authors. As suggested by another reviewer, a functional study using FEA might be possible to test this assumption. I am not suggesting that his should be done in this study, as further fossil taxa and extant analogues would be necessary to test head-butting comprehensively and with some confidence. However, this could be mentioned as possibility for future studies.

·

Basic reporting

The genus name Stahleckeria is misspelled throughout the text.

Experimental design

No comment.

Validity of the findings

I do not find the arguments that "this is the first scan of tapinocephalids" convincing as to why no broader anatomical or phylogenetic context could be included (and indeed view such context as vital for evaluating the proposals herein), but guess I will have to agree to disagree with the authors on this point.

The reply of "no comment" to the factual statement that the combat structures of ungulates are commonly preserved in the fossil record is insufficient.

---

## Round 0.3 · Minor Revisions

I note you did not deal with the question raised by Reviewer #2 regarding the "broader anatomical or phylogenetic context" nor have you acknowledged the work of the three reviewers. May I suggest you address these two points or, alternatively, give some explanation why at least the former is undesirable in your opinion (of course, it is your own choice who to acknowledge)?

Also, please note that your manuscript cannot be accepted before we can confirm that the original dataset has been made publicly accessible. I would be grateful if you could provide the exact url of the webpage (when activated) from where the dataset can be downloaded (not the general url of the “ESRF webstite [sic]”, where I could not find your data to download).

---

## Round 0.4 · accepted · Accept

Your manuscript has now been accepted for publication. Congratulations!

Please, fix the typos and improper spelling, grammar and punctuation (e.g., lines 72, 134, 135, 154, 381, 420, 422, 457, 743, 776, 809) when you get the chance.